# RAPNet: Accelerating Algebraic Multigrid with Learned Sparse Corrections

**Yali Fink** [* 1]  **Ido Ben-Yair** [* 1]  **Lars Ruthotto** [2]  **Eran Treister** [1]

## Abstract

The scalable solution of large sparse linear systems is a bottleneck in scientific computing and graph analysis. While algebraic multigrid (AMG) offers optimal linear scaling, its performance is severely constrained by the trade-off between the sparsity and convergence quality of coarse-grid operators. Classical AMG heuristics struggle to balance these objectives, often sacrificing stability or performance for sparsity. We propose RAPNet, a graph neural network (GNN) framework that resolves this trade-off by learning to generate sparse, robust coarse operators directly from the sparse algebraic system. Key to our approach is a level-wise training strategy that enables learning from small subgraphs and generalization to million-node domains, bypassing the bottlenecks of prior neural AMG attempts. RAPNet executes exclusively during the solver setup phase, ensuring that the solve phase retains its favorable computational properties. We show that our method outperforms classical non-Galerkin baselines on diverse PDE discretizations and graph Laplacians, making it particularly effective for multi-query tasks such as eigenproblems, time-dependent simulations, and inverse or design problems.

## 1. Introduction

Solving sparse linear systems $A\mathbf{x} = \mathbf{b}$ is a fundamental computational primitive in machine learning and scientific computing, arising in spectral graph analysis, PDE-based simulation, and large-scale optimization. Iterative solvers with effective preconditioners become essential as problems scale to millions of unknowns (Spielman & Teng, 2014; Stüben, 2001). Designing high-quality preconditioners requires domain expertise and careful tuning, motivating the recent surge in machine learning approaches.

While several neural acceleration paradigms have been shown to be effective, they also face limitations compared to traditional methods. *Neural surrogates* (Li et al., 2021; Lu et al., 2021) learn to map $\mathbf{b}$ directly to $\mathbf{x}$. Yet, they provide fixed-accuracy estimates bounded by network capacity and often struggle with out-of-distribution inputs. Conversely, *operator-based* paradigms directly attempt to learn $M \approx A^{-1}$ or its sparse factors. However, these often require costly neural inference at every solver iteration. Furthermore, theory by Trifonov et al. (2026) shows that message-passing GNNs fundamentally cannot approximate sparse triangular factorizations for broad classes of unstructured problems.

Alternatively, performance can be improved by *augmenting* classical solvers such as algebraic multigrid (AMG) or matrix factorizations. This can often be done while retaining some of their desirable properties (Greenfeld et al., 2019; Luz et al., 2020; Trifonov et al., 2025). Unfortunately, prior methods often struggle to scale to large systems or deep hierarchies (Luz et al., 2020; Chen, 2025). In addition, they typically sacrifice sparsity, require per-iteration inference, or deal only with a fixed $\mathbf{b}$. Even recent neural preconditioning operators (Li et al., 2025) rely on soft attention mechanisms that effectively act as dense operators.

To date, no existing approach combines *strict sparsity* by construction, *efficient scaling*, and *setup-only inference*. We fill this gap with **RAPNet**—a GNN framework that learns to emit sparse additive corrections to sparse AMG operators. Because AMG offers optimal linear complexity (Xu & Zikatanov, 2017) and exposes parallelism when coarse operators are sufficiently sparse (De Sterck et al., 2006; Bienz et al., 2016), AMG provides an ideal skeleton for augmentation. Drawing on classical methods (Falgout & Schroder, 2014; Treister & Yavneh, 2015), RAPNet executes *only once* during the setup of the solver. The subsequent solve phase applies AMG iteratively using efficient vector and sparse matrix operations. To scale to massive linear systems, RAPNet utilizes a shared-weight architecture that exploits the spatial locality of multigrid operators, allowing efficient

---

*Equal contribution   [1] Institute for Interdisciplinary Computational Sciences, Faculty of Computer and Information Science, Ben-Gurion University of the Negev, Be'er Sheva, Israel [2] Department of Mathematics and Computer Science, Emory University, Atlanta, GA, USA. Correspondence to: Ido Ben-Yair <idobeny@post.bgu.ac.il>.

*Proceedings of the 43$^{rd}$ International Conference on Machine Learning*, Seoul, South Korea. PMLR 306, 2026. Copyright 2026 by the author(s).

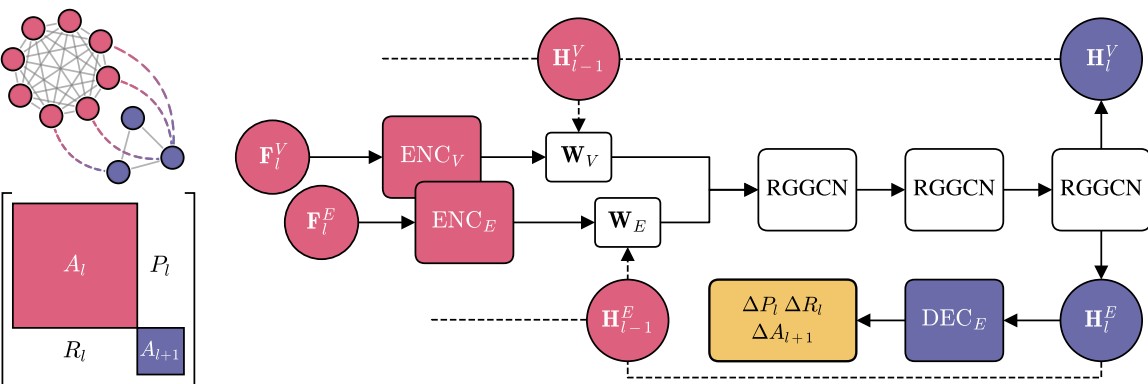

*Figure 1.* **The RAPNet architecture.** The model processes the AMG hierarchy as a sequence of level pairs. **(Left)** We model the fine ($l$) and coarse ($l + 1$) levels as a single composite graph, where transfer operators ($P_l$, $R_l$) act as inter-level edges. This forms a block system matrix, which is converted to input features (see Equations (5) to (7)). **(Right)** Using a weight-shared architecture, the model propagates messages across this topology to predict sparse corrections for $A_{l+1}$, $P_l$, and $R_l$. The hidden state is mixed with the next level pair ($l \leftarrow l + 1$) to capture hierarchical dependencies, applying the same learned parameters throughout the entire hierarchy.

training on small graphs without sacrificing generalization.

Our key contributions are:

- augmenting AMG operators exclusively during the setup phase, accommodating any number of right-hand sides,

- scaling to large-scale problems with deeper hierarchies in inference than those seen in training, achieved through a level-wise, shared-weight architecture, and

- training efficiently while generalizing to million-node systems by employing multiple techniques to reduce the sizes of training graphs.

We evaluate the V-cycles that RAPNet generates as standalone solvers and preconditioners, showing significantly reduced iteration counts over classical AMG variants. The architecture is visualized in Figure 1.

## 2. Related Work

Using machine learning to accelerate linear solvers is a currently active field. Recent advances have spurred interest in replacing solvers entirely with learning-based methods. These approaches generally fall into two categories: discrete end-to-end models, i.e., *neural surrogates*, and continuous end-to-end models, often called *neural operators*.

Seminal *surrogate* papers by Pfaff et al. (2021) and Sanchez-Gonzalez et al. (2020) address mesh- and particle-based dynamics for fluid mechanics respectively. More recently, Lam et al. (2023) applied GNNs to global weather forecasting. These methods offer geometric flexibility but require engineering to a specific PDE. They are also often limited by

the receptive field of global message passing. Others upscale coarse-grid solutions from differentiable solvers with GNNs (Belbute-Peres et al., 2020; Fortunato et al., 2022; Horie & Mitsume, 2022), which requires neural inference during the solve.

Neural *operators* aim to learn mappings between infinite-dimensional function spaces. Prominent architectures include DeepONets (Lu et al., 2021), Fourier neural operators (Li et al., 2021), and graph neural operators (Kovachki et al., 2023). These rely on the fast Fourier transform (FFT), global message passing, or coordinate transformations (Li et al., 2023) to apply the FFT, but this approach fails when the graph cannot be modeled well by a diffeomorphic coordinate transformation, such as power-law or social network graphs. More recent methods combine multigrid methods with neural operators to conserve their properties, but require inference during the solve (He et al., 2024; Hu et al., 2025), can be difficult to train (Rudikov et al., 2024) or use the self-attention mechanism (Li et al., 2025). These methods function as statistical surrogates rather than numerical solvers, and do not allow for the iterative reduction of the residual and error. Others have attempted to bridge the gap between operators and numerical solvers such as Krylov methods (Zhang et al., 2024; Kopaničáková et al., 2025; Kopaničáková & Karniadakis, 2025). However, they typically apply the neural network at each iteration of the solution process.

Neural *Krylov solvers* have also been considered. Chen (2025) showed that GNNs can effectively capture the spectral properties of a black-box preconditioner, while others approximate matrix inverses or factorizations (Häusner et al., 2024; Trifonov et al., 2025; Häusner et al., 2025). Similar CNN-based methods also exist, where the network acts as

the preconditioner (Azulay & Treister, 2023; Huang et al., 2022; Lerer et al., 2024; Han et al., 2024).

Other approaches include using neural networks to learn tuning parameters for classical algorithms (Antonietti et al., 2023; Caldana et al., 2024; Antonietti et al., 2026).

This work builds on the papers by Greenfeld et al. (2019) and Luz et al. (2020). These methods learn effective AMG prolongation operators directly from the system matrix. However, they were designed for two levels only, limiting their scalability. More recently, Huang et al. (2024) proposed a neural approach to sparsifying coarse operators with two neural networks to predict the sparsity and values of the coarse stencils, respectively. However, they rely on convolutions in structured domains. Similarly, Taghibakhshi et al. (2023) use a GNN to improve domain-decomposition solvers. Our method overcomes these limitations, enabling generalization to multi-level hierarchies and larger, unstructured domains.

## 3. AMG Background

AMG algorithms are efficient for solving certain types of linear systems, primarily those arising from discretizations of elliptic PDEs or similarly behaving problems (Ruge & Stüben, 1987; Brandt, 1986; Stüben, 2001). For these problems, traditional methods like Jacobi or even plain Krylov methods tend to converge slowly due to the presence of long-range error that corresponds to relatively small eigenvalues, i.e., low-frequency error. AMG uses lower-resolution, i.e., coarser, representations of $A$ to communicate information across different scales, thereby connecting distant unknowns to achieve rapid convergence. This is because such low-frequency error is slow to eliminate by smoothing, yet it is eliminated naturally when projected onto coarser scales: at coarser scales, it appears as high-frequency error, where smoothers eliminate it quickly. Since this error already appears smooth to the fine smoothers, it is known as *algebraically-smooth*. Note that it is the eigenbasis of $A$ that determines the frequencies and smoothness under consideration, not necessarily the standard Fourier eigenbasis, hence the term *algebraic*. AMG methods are used either as standalone solvers or as preconditioners for Krylov methods.

Starting from the finest level where $A_0 = A \in \mathbb{R}^{n \times n}$, AMG defines coarse level operators $A_1, A_2, \ldots, A_L$ ($A_l \in \mathbb{R}^{n_l \times n_l}$, $n_l > n_{l+1}$) by the Galerkin product with the *transfer operators*:

$$A_{l+1} = R_l A_l P_l \in \mathbb{R}^{n_{l+1} \times n_{l+1}}, \quad (1)$$

where $P_l \in \mathbb{R}^{n_l \times n_{l+1}}$ is the *prolongation* operator interpolating vectors from level $l+1$ to level $l$, and $R_l \in \mathbb{R}^{n_{l+1} \times n_l}$ is the *restriction* operator from level $l$ to level $l+1$. The

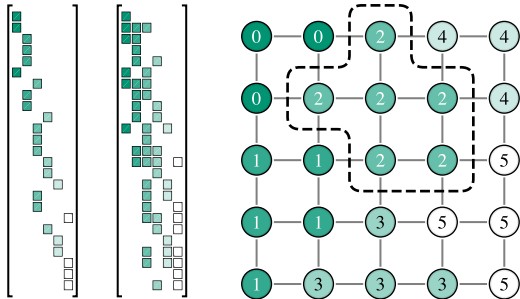

*Figure 2.* **Aggregation and stencil growth. (Right)** A grid graph decomposed into disjoint aggregates (colored). **(Left)** The sparsity pattern of the unsmoothed prolongation $P$. It only has a single non-zero entry per row, corresponding to the aggregate assignment of the node. **(Middle)** The smoothed prolongation operator. Note the significant increase in density, as the smoothing operation causes fine nodes to interpolate from multiple coarse nodes.

choice of transfer operators $P_l$ and $R_l$ crucially impacts the convergence rate of the solver, and thus choosing them effectively is central in AMG research. See Section A for a detailed introduction to multigrid methods.

The AMG two-level error iteration is given by

$$\mathbf{e}^{(k+1)} = S^{\nu_2}(I - PA_g^{-1}RA)S^{\nu_1}\mathbf{e}^{(k)}, \quad (2)$$

where $\mathbf{e}^{(k)}$ is the error at the $k$-th iteration, $S^{\nu_1}$ is the pre-smoother (e.g., Jacobi) applied $\nu_1$ times, and $S^{\nu_2}$ denotes the post-smoother applied $\nu_2$ times. $A_g$ is the Galerkin coarse-grid operator, a specific instantiation of $A_{l+1}$ in (1). When applying more than two levels, the coarse-grid solve $A_g^{-1}$ is replaced by a recursive application of (2). Galerkin AMG works by first smoothing the error with $S^{\nu_1}$. Then, the coarse level of the multigrid cycle, i.e., $A_g$ in (2) eliminates the remaining error, which is now smooth.

One way to build $P$ and $R$ is to assume local piecewise-constant error, which corresponds to forming disjoint aggregates of neighboring nodes where error is constant within each such aggregate (see Figure 2). This approach creates highly sparse $P$ and $R$ matrices, but for many systems, it fails to capture the coarse error well and often leads to slow convergence. Heuristics such as smoothed aggregation (SA) can address this (Vaněk et al., 1996). SA applies a smoothing operation to the piecewise-constant interpolator. While this often improves convergence, smoothing $P$ and $R$ dramatically increases their density. This phenomenon, called *stencil growth*, imposes computational costs and communication issues in parallel settings (De Sterck et al., 2006; Bienz et al., 2016; Huang et al., 2024).

Using a coarse-grid operator other than $A_g$ is known as non-Galerkin AMG. Such methods find a sparser $A_c$ such that

$A_c$ is spectrally-equivalent to $A_g$, i.e.,

$$A_g \mathbf{e}_c \approx A_c \mathbf{e}_c \qquad (3)$$

for any smooth coarse error vector $\mathbf{e}_c$ that $A_c$ will encounter if substituted into (2). Simply put, the difference in action between $A_g$ and $A_c$ is minimized, while $A_c$ is sparser than $A_g$ (Falgout & Schroder, 2014; Treister & Yavneh, 2015; Bienz et al., 2016). For example, consider the 5-point Laplacian stencil for a regular grid, which results in a 9-point Galerkin operator $A_g$ for which the 5-point Laplacian operator $A_c$ is spectrally equivalent:

$$A_g : \frac{1}{64} \begin{bmatrix} -1 & -2 & -1 \\ -2 & 12 & -2 \\ -1 & -2 & -1 \end{bmatrix} A_c : \frac{1}{16} \begin{bmatrix} & -1 & \\ -1 & 4 & -1 \\ & -1 & \end{bmatrix} . \quad (4)$$

The equivalence (3) cannot hold for every $\mathbf{e}_c$ (Falgout & Schroder, 2014), and consequently, these methods may struggle to preserve the spectral properties required for convergence. Moreover, they often rely on the construction of the denser Galerkin operator $RAP$ prior to sparsification, and the identification of a high-quality coarse operator to begin with. This density-convergence trade-off prompts us to apply machine learning to this problem, avoiding computing the denser operator while achieving convergence rates similar to smoothed aggregation.

Approximations exist for other stencil operators for regular grids (Bolten & Frommer, 2007; Bolten et al., 2016). However, analytically deriving superior operators for non-regular grids is difficult. Hence, in this work, we leave the sparsity pattern given by the unsmoothed aggregation for $A_g$, i.e., we set $\mathrm{sp}(A_c) = \mathrm{sp}(A_g)$, which is guaranteed to be sufficiently sparse under mild assumptions. We only improve the values of $A_c$. This ensures that the operator complexity remains near the theoretical minimum of plain unsmoothed aggregation while capturing the performance benefits of a high-quality smoothed hierarchy.

In summary, if our GNN can construct coarse operators that are spectrally equivalent for algebraically-smooth error modes—*without computing the denser Galerkin operators*—then both efficiency and rapid convergence can be achieved.

### 3.1. Classical Sparsification

Classical sparsification methods improve convergence at the cost of fundamental algorithmic bottlenecks that impede scalability. Sparsification, i.e., computing the denser smoothed operators and then removing entries, effectively computes all distance-2 and distance-3 neighborhoods of each node. To understand why, consider that the Galerkin product produces overlapping partial sums

$$[RAP]_{ij} = \sum_k \sum_l R_{ik} A_{kl} P_{lj}.$$

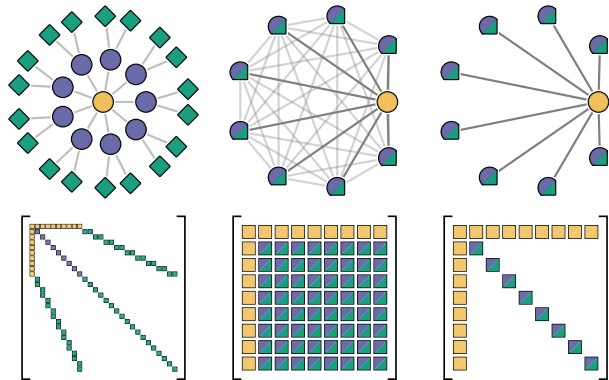

*Figure 3.* **Sparsity-density trade-off.** A comparison of sparsity patterns for the graph Laplacian. **(Left)** The original fine-level operator. **(Middle)** The coarse operator generated by smoothed aggregation. Note the increased density. **(Right)** The unsmoothed aggregation coarse operator with minimal sparsity.

If $P$ and $R$ are smoothed, this sums up the values for neighborhoods up to distance 3 due to the action of $R$, then $A$, then $P$ on each node. This scales poorly with the average degree of each node, which corresponds to density. Moreover, in a sparse-sparse product, many of these entries are computed separately in parallel, and then a sorting and summation stage yields the final values. This exposes a fundamental paradox in sparsification: it is **algorithmically as dense as the smoothed Galerkin product**. In contrast, RAPNet is structurally static, bypassing the calculation of the smoothed operators by message passing on the unsmoothed hierarchies. Figure 3 illustrates the stencil growth in sparsified smoothed aggregation (SpSA) (Treister & Yavneh, 2015), which we consider as a representative of sparsification methods here.

## 4. RAPNet

RAPNet retains the efficient sparsity pattern of simple aggregation but learns additive corrections to improve convergence. Classical aggregation heuristics provide a strong, albeit imperfect, inductive bias for the multigrid hierarchy. Leveraging this, we initialize the hierarchy using standard piecewise-constant unsmoothed aggregation operators (denoted $P_g^{(l)}$ and $R_g^{(l)}$ for level $l$), which contain binary aggregate selector weights. Our GNN then predicts additive corrections $\Delta P$, $\Delta R$, and $\Delta A_c$ constrained to the non-zero entries defined by the aggregation pattern. As we show in Section 5, this results in a convergence rate similar to or exceeding that of classical baselines with a negligible cost post-training.

Training deep learning models on high-dimensional graph data presents significant memory and computational bottle-

necks. To scale training to large graphs, we exploit the local nature of the problems at hand using multiple techniques. We show that RAPNet is capable of generalizing from 2D problems seen in training to 3D problems during evaluation and scale to large graphs.

We address the challenge of scaling to deep multigrid hierarchies through a *level-wise training curriculum* that exploits the recursive nature of the algebraic multigrid hierarchy. This restricts the GNN to adjacent levels of the multigrid hierarchy, pairing one fine level with its immediate coarse representation, rather than unrolling the full cycle.

The rest of this section details the design of RAPNet and how it achieves these goals.

## 4.1. Design and Architecture

RAPNet processes the AMG hierarchy as a sequence of level pairs $(l, l+1)$. Each pair is represented as a composite graph where the fine- and coarse-level variables serve as nodes. The off-diagonals of $A_l$ and $A_{l+1}$ are intra-level edges, while the transfer operators $P_{\mathrm{g}}^{(l)}$ and $R_{\mathrm{g}}^{(l)}$ serve as inter-level edges. This topology enables simultaneous message passing both within and between grid resolutions. A weight-shared encoder-processor-decoder architecture, illustrated in Figure 1, is applied recursively to every level pair.

The GNN architecture itself follows the encode-process-decode paradigm. The encoders map initial node and edge attributes onto a latent space; the processor propagates information by message-passing; and the decoder projects the refined latent representations back to the physical domain to predict operator corrections. This sequential design promotes scale-invariance by encouraging the network to learn local dynamics that are independent of the global hierarchy depth. This enables it to learn using only a few levels and then generalize to more levels in inference.

**Composite graph construction** For each level $l$, the composite graph $\mathcal{G}_l = (\mathcal{V}_l, \mathcal{E}_l)$ represents the transition between the fine level $l$ and the coarse level $l + 1$. Let $n_l$ and $n_{l+1}$ denote the number of nodes at each respective level, and let $\mathcal{V}_l \cup \mathcal{V}_{l+1}$ be the composite node set of size $N = n_l + n_{l+1}$. The composite graph is represented by a block matrix $\mathbf{B}_l \in \mathbb{R}^{N \times N}$ illustrated in Figure 1

$$\mathbf{B}_l = \begin{bmatrix} A_l & P_l \\ R_l & A_{l+1} \end{bmatrix}, \tag{5}$$

where $A_l$ and $A_{l+1}$ are the system operators, and $P_l \in \mathbb{R}^{n_l \times n_{l+1}}$ and $R_l \in \mathbb{R}^{n_{l+1} \times n_l}$ are the transfer operators. These inter-level edges are shown as dashed lines in Figure 1. This weighted adjacency matrix $\mathbf{B}_l$ enables message-passing operations to simultaneously capture intra-level

smoothing (via $A_l, A_{l+1}$) and inter-level corrections (via $P_l, R_l$) during each step of the hierarchical training.

**Node and edge features** To distinguish between levels, we define the node features $\mathbf{F}_l^V \in \mathbb{R}^{N \times 2}$ as a concatenation of one-hot vectors corresponding to the fine and coarse nodes, respectively

$$\mathbf{F}_l^V = \begin{bmatrix} \mathbf{1}_{n_l} & \mathbf{0}_{n_l} \\ \mathbf{0}_{n_{l+1}} & \mathbf{1}_{n_{l+1}} \end{bmatrix}. \tag{6}$$

The edge feature matrix $\mathbf{F}_l^E \in \mathbb{R}^{|\mathcal{E}_l| \times 5}$ encodes the scalar weights and structural role of each edge

$$\mathbf{F}_l^E = \begin{bmatrix} \mathrm{vals}\,(\mathbf{B}_l) & \mathbb{I}_{A_l} & \mathbb{I}_{P_l} & \mathbb{I}_{R_l} & \mathbb{I}_{A_{l+1}} \end{bmatrix}, \tag{7}$$

where $\mathrm{vals}\,(\mathbf{B}_l)$ are the non-zero entry values, and $\mathbb{I}_{(\cdot)}$ are binary vectors of length $|\mathcal{E}_l|$ indicating whether an edge belongs to the respective component $A_l, P_l, R_l$, or $A_{l+1}$.

**Encoders** For each pair of levels, the *learned encoders* project the feature matrices onto a high-dimensional latent space. This initializes the latent node and edge states, $\mathbf{H}_l^V \in \mathbb{R}^{N \times 64}$ and $\mathbf{H}_l^E \in \mathbb{R}^{|\mathcal{E}_l| \times 64}$

$$\mathbf{H}_l^V = \mathrm{ENC}_V\left(\mathbf{F}_l^V\right), \quad \mathbf{H}_l^E = \mathrm{ENC}_E\left(\mathbf{F}_l^E\right), \tag{8}$$

where $\mathrm{ENC}_V$ and $\mathrm{ENC}_E$ are learned MLP sequences applied to the feature matrices. These latent representations are then fed to the subsequent message-passing layers in the processor.

Let $\mathbf{H}_{l-1}^V[c]$ denote the final coarse node latent states from the previous iteration, and $\mathbf{H}_l^V[f]$ denote the initial encoded fine node states in the current iteration. We update the fine embeddings via a concatenation and a *learned* linear projection $\mathbf{W}_V$, i.e.,

$$\mathbf{H}_l^V[f] \leftarrow \left[\mathbf{H}_l^V[f] \,\middle\|\, \mathbf{H}_{(l-1)}^V[c]\right] \cdot \mathbf{W}_V. \tag{9}$$

A similar *learned* linear mixing operation, denoted $\mathbf{W}_E$, is applied to the edge embeddings. This mechanism allows the network to act as a deep recurrent unit over the depth of the multigrid hierarchy.

**Processor** Following the encoding and mixing steps, the latent graph representation undergoes message passing using layers of residual gated graph convolutional nets (RGGCN) (Bresson & Laurent, 2018), which also contain learnable parameters. Each RGGCN layer updates the node and edge representations, $\mathbf{H}_l^V$ and $\mathbf{H}_l^E$, respectively, by integrating local neighborhood information. We selected this message-passing layer due to its ability to process anisotropic information required to learn anisotropic local dynamics. Residual connections within the processor reduce the oversmoothing effect common in deep GNNs.

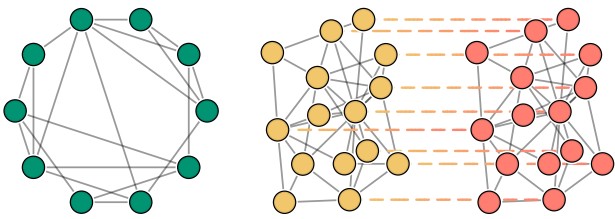

*Figure 4.* **Graph dataset examples. (Left)** A Watts-Strogatz graph. **(Right)** A temporal Barabási-Albert graph.

**Decoder**  After processing, the final edge embeddings $\mathbf{H}_l^E$ are projected back to the scalar domain to predict the correction values for the operators $P_l$, $R_l$ and $A_{l+1}$:

$$\Delta P_l,\ \Delta R_l,\ \Delta A_{l+1} = \text{DEC}_E\left(\mathbf{H}_l^E\right). \qquad (10)$$

where the decoder, $\text{DEC}_E$, is also a *learned* MLP sequence.

### 4.2. Training

Our training scheme aims to yield models that can operate on entire classes of matrices of a certain type, not only single instances of a class. For example, illustrative examples of two of the datasets are presented in Figure 4. Accordingly, we design the solver to solve the system $A\mathbf{x} = \mathbf{b}$ for any right-hand side $\mathbf{b}$. Our datasets do not include any specific right-hand sides. Instead, we generate synthetic right-hand side instances during training, so the network learns to handle any right-hand side $\mathbf{b}$.

**Batched residual generation**  We train RAPNet in a self-supervised manner. For each training step, we sample a sparse matrix $A \in \mathbb{R}^{n \times n}$ and its AMG hierarchy from our dataset. To ensure gradient stability, we generate a batch of $n_b$ independent ground-truth solution vectors $\{\mathbf{x}_{\text{true}}^{(i)}\}_{i=1}^{n_b}$, where each $\mathbf{x}_{\text{true}}^{(i)} \sim \mathcal{N}(0, I)$. The corresponding batch of right-hand sides is then computed as $\mathbf{b}^{(i)} = A\mathbf{x}_{\text{true}}^{(i)}$.

The randomly generated solutions tend to have high-frequency content, which is eliminated efficiently by the classical smoothers. To target low-frequency, algebraically-smooth error modes, we avoid training RAPNet on random noise directly. Instead, we generate an initial guess $\mathbf{x}_0^{(i)}$ by applying a random number of standard AMG V-cycles $k_i \sim \mathcal{U}(1, 30)$ to a random initialization. The 30-iteration range was determined experimentally. We observed that solvers begin to stall on most of our datasets starting at the 15-iteration mark once high-frequency modes are fully damped. This yields the residual and error pair

$$\mathbf{r}^{(i)} = \mathbf{b}^{(i)} - A\mathbf{x}_{k_i}^{(i)}, \quad \mathbf{e}^{(i)} = \mathbf{x}_{\text{true}}^{(i)} - \mathbf{x}_{k_i}^{(i)}, \qquad (11)$$

which satisfy the fundamental error-residual relationship

$$A\mathbf{e}^{(i)} = \mathbf{r}^{(i)}. \qquad (12)$$

In this formulation, the residual $\mathbf{r}^{(i)}$ serves as the right-hand side for the error correction problem.

**Differentiable cycle and loss**  RAPNet processes the hierarchy of $A$ to predict an additive correction for each AMG operator. We augment the V-cycle using the corrections emitted by RAPNet, then perform a single V-cycle using the augmented hierarchy to solve (12) and obtain an error vector $\hat{\mathbf{e}}^{(i)}$. Since the inverse coarse matrix is often ill-conditioned and can be prone to instability, we replace the exact coarse solve during training with two iterations of Jacobi smoothing. Experimentally, we found that this promotes gradient stability during training. This approach stabilizes the end-to-end training process and avoids the overhead of a direct solver on a GPU.

RAPNet is trained to minimize the discrepancy between the predicted error $\hat{\mathbf{e}}^{(i)}$, and the true error $\mathbf{e}^{(i)}$ using a squared relative $\ell_2$ loss to preserve scale-invariance across different error scales

$$\mathcal{L} = \frac{1}{n_b} \sum_{i=1}^{n_b} \frac{\|\hat{\mathbf{e}}^{(i)} - \mathbf{e}^{(i)}\|_2^2}{\|\mathbf{e}^{(i)}\|_2^2 + \epsilon}, \qquad (13)$$

where $\epsilon = 10^{-12}$ is a small constant for numerical stability. By optimizing this objective over a batch of right-hand sides, RAPNet learns to construct hierarchies that are robust to different levels of smoothness of the residuals. We note that due to the smoothing of the initial data, the initial distribution of the solution vectors becomes irrelevant since the network only sees algebraically-smooth error vectors. This smoothness is determined solely by $A$ and the chosen smoother.

**Subgraph training**  Since RAPNet can accommodate graphs of any size due to its shared weights, we train the network on **small graphs** to learn local dynamics between the fine, coarse, and transfer operators. As commonly done using GNNs for other tasks (Yehudai et al., 2021), we apply RAPNet to **much larger** problems during testing, which also requires using more levels in the AMG hierarchy. This is a unique aspect of this work in the context of solving linear systems. The reason why this is possible may be explained by common multigrid theory: multigrid performance is often analyzed through local Fourier analysis (LFA) or similar multiscale-local analysis methods. We extract patches or generate smaller similar problems with the intention of mimicking the smooth behavior of the solution beyond the patch: we train the anisotropic and advection-diffusion models using small 2D domains, and evaluate on large 3D domains; for graph Laplacians, we train by generating large graphs and extracting smaller graphs from them. We extract submatrices out of these large adjacency matrices and then form the Laplacians for these submatrices. This principle is also sometimes used for constructing basis functions to gener-

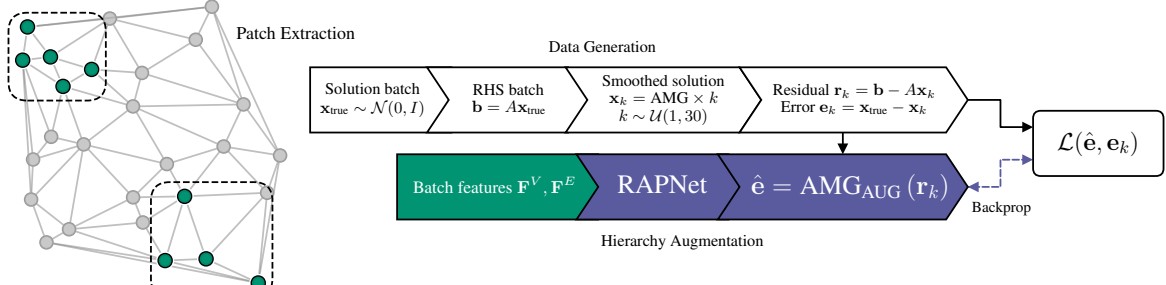

*Figure 5.* **RAPNet training pipeline. (Left)** Input preparation. To ensure generalization to large systems, we train on local patches extracted from larger graphs or ones generated as small variants. **(Right)** We generate training data by applying a random number of AMG cycles to ground-truth vectors, producing algebraically-smooth residuals $\mathbf{r}_k$ and errors $\mathbf{e}_k$. This demanding generation step is gradient-free; backpropagation is only performed through the RAPNet prediction and the final augmented V-cycle application.

ate interpolations based on local graph patches (Chartier et al., 2003). The full subgraph extraction procedure is given in Section B.2. The residual generation and training process are depicted in Figure 5.

**Inference and Generalization**   We train RAPNet to augment a Galerkin AMG hierarchy. The resulting hierarchy is therefore a non-Galerkin hierarchy. Non-Galerkin theory (Falgout & Schroder, 2014) states that if the gap $\varepsilon = \|A_g - A_c\|_F$ is small, then the non-Galerkin cycle will behave similarly to the Galerkin cycle. By construction, this also holds for RAPNet because the cycle we apply in inference is a standard AMG cycle, regardless of the fact that its non-zero values are modified by our GNN (although RAPNet is not constrained to small perturbations).

There are two key differences between training and evaluation. First, training uses smaller domains and fewer levels than evaluation. Second, the training objective (13) considers a single cycle to reduce computational costs, while evaluation applies the cycle iteratively until convergence, either as a solver or preconditioner. This avoids the computational cost of unrolling multiple cycles during training, which would require backpropagation through the entire unrolled sequence.

**Limitations**   We note a few limitations of RAPNet here. One such limitation is that the smoothing process within the AMG V-cycle, i.e., damped Jacobi, remains purely classical. That is, its damping parameters or operators are not corrected by RAPNet. Second, RAPNet uses classical neighborhood aggregation algorithms and does not learn to aggregate the nodes. This can in principle be learned via node clustering GNNs. We leave the exploration of these features for future work. Lastly, RAPNet lacks theoretical guarantees for the preconditioners it generates. Note that non-Galerkin theory only provides convergence guarantees for small perturbations, which often limits the possible

improvement to the solver in practice. RAPNet does not restrict its corrections to this regime, and its improvements are empirically observed rather than theoretically guaranteed.

## 5. Numerical Experiments

RAPNet is explicitly intended to overcome the limitations inherent in SpSA (Treister & Yavneh, 2015), while preserving a lightweight setup process similar to plain aggregation, denoted by AGG in this section. Thus, we evaluate the performance of RAPNet in comparison to the classical baselines AGG and SpSA across a diverse suite of linear systems. SpSA also represents the approaches of (Falgout & Schroder, 2014; Bienz et al., 2016), and is considered quite robust and powerful in this setting. We used common default parameters for the classical methods. The AGG hierarchies and the inputs to RAPNet employ the same parameters. This ensures that better iteration counts for RAPNet strictly improve over the corresponding AGG experiment. The parameter values that are used are detailed in Section B.

To the best of our knowledge, no current learned method is capable of scaling past two multigrid levels, or to comparable problem sizes as RAPNet. Therefore, we do not consider any learned methods for comparison. One notable exception is by Chen (2025), which shows the method can scale up to 100K nodes. However, it trains a separate network for each instance of the problem, whereas RAPNet trains once for an entire dataset of similar linear systems. Hence, we did not consider it to be an apt comparison.

We evaluate each method as a standalone solver, and as a preconditioner to GMRES (Saad, 1993). In both cases, we measure performance by the number of iterations required to reach a relative residual of $10^{-6}$ on a random right-hand side. This threshold approaches the numerical precision limits of the single-precision arithmetic that was used in all experiments. The rest of this section details our experimen-

*Table 1.* Average iteration counts for AGG, SpSA, and RAPNet. Each experiment represents the average of 100 different examples with a limit of 1000 iterations. Performance is given as standalone solvers and as preconditioners for GMRES(2) across different benchmarks. The notation – indicates non-convergence on the entire test dataset. Note that large standard deviations may stem from exhausting the iteration limit on some examples. See Table 12 for individual convergence rates.

| Experiment | Vars | NNZ | As Standalone Solver | | | As Preconditioner to GMRES | | |
|---|---|---|---|---|---|---|---|---|
| | | | AGG | SpSA | RAPNet | AGG | SpSA | RAPNet |
| 2D Geometric | 128K | 896K | 160 ± 7 | 100 ± 117 | **83 ± 4** | 48 ± 2 | **36 ± 2** | **36 ± 2** |
| (Symmetric) | 256K | 1.8M | 161 ± 5 | 141 ± 200 | **83 ± 3** | 48 ± 1 | **36 ± 1** | 37 ± 1 |
| | 512K | 3.6M | 163 ± 3 | 108 ± 132 | **84 ± 2** | 48 ± 1 | **36 ± 1** | 37 ± 1 |
| | 1M | 7M | 163 ± 2 | 148 ± 203 | **84 ± 1** | 49 ± 1 | **36 ± 1** | 37 ± 1 |
| 3D Geometric w/ | 128K | 2.1M | 132 ± 17 | 63 ± 10 | **59 ± 5** | 39 ± 4 | **30 ± 2** | **30 ± 8** |
| Random Weights | 256K | 4.2M | 142 ± 16 | **64 ± 8** | **64 ± 6** | 41 ± 4 | **30 ± 2** | 31 ± 2 |
| (Asymmetric) | 512K | 8.4M | 145 ± 18 | **65 ± 8** | 71 ± 6 | 42 ± 5 | **32 ± 2** | 34 ± 2 |
| Watts-Strogatz w/ | 128K | 896K | 419 ± 154 | 513 ± 412 | **302 ± 91** | 462 ± 442 | 303 ± 405 | **109 ± 24** |
| Random Weights | 256K | 1.8M | 409 ± 105 | 547 ± 408 | **299 ± 85** | 501 ± 445 | 355 ± 425 | **119 ± 31** |
| (Asymmetric) | 512K | 3.6M | 446 ± 135 | 689 ± 380 | **317 ± 59** | 472 ± 434 | 450 ± 452 | **123 ± 22** |
| | 1M | 7M | 453 ± 136 | – | **325 ± 69** | 549 ± 439 | 414 ± 383 | **130 ± 20** |
| Temporal | 200K | 1.4M | 102 ± 22 | 467 ± 120 | **72 ± 21** | **28 ± 4** | 97 ± 64 | 38 ± 9 |
| Barabási-Albert | 400K | 2.8M | 104 ± 16 | 216 ± 228 | **74 ± 16** | **28 ± 3** | 44 ± 51 | 41 ± 8 |
| (Symmetric) | 600K | 4.2M | 106 ± 13 | 90 ± 152 | **85 ± 45** | 28 ± 3 | **25 ± 25** | 44 ± 7 |
| Social Hub | 100K | 1M | 43 ± 14 | 500 ± 493 | **19 ± 2** | 14 ± 3 | 16 ± 6 | **10 ± 1** |
| (Symmetric) | 200K | 2.1M | 34 ± 11 | 528 ± 494 | **17 ± 2** | 13 ± 2 | 16 ± 8 | **10 ± 1** |
| | 300K | 3.1M | 61 ± 166 | 978 ± 115 | **18 ± 12** | 13 ± 4 | 34 ± 94 | **9 ± 1** |
| 3D FEM | 227K | 2.7M | 228 ± 21 | **132 ± 10** | 202 ± 17 | 63 ± 4 | **60 ± 4** | 59 ± 4 |
| Anisotropic Diffusion | 355K | 4.3M | 262 ± 25 | **151 ± 12** | 231 ± 20 | 70 ± 5 | **66 ± 4** | 65 ± 4 |
| (Symmetric) | 524K | 6.5M | 285 ± 29 | **164 ± 14** | 251 ± 24 | 74 ± 6 | 71 ± 5 | **69 ± 5** |
| 3D FEM | 227K | 2.7M | – | 971 ± 165 | **73 ± 18** | 79 ± 21 | 45 ± 9 | **38 ± 4** |
| Advection-Diffusion | 355K | 4.3M | 865 ± 336 | 791 ± 397 | **61 ± 19** | 80 ± 96 | 41 ± 11 | **36 ± 6** |
| (Asymmetric) | 524K | 6.5M | 906 ± 285 | 875 ± 324 | **62 ± 15** | 80 ± 28 | 42 ± 9 | **38 ± 5** |

tal setup and the specifics of each experiment.

**Experimental Setup** Our experiments cover graph Laplacians and discretized PDEs. For graph Laplacians, we evaluated RAPNet on multiple datasets of graphs. The experiments denoted as "geometric" are generated by a Delaunay triangulation of random points on the unit square or cube. Next, we use Watts-Strogatz graphs (Watts & Strogatz, 1998) and temporal Barabási-Albert (TBA) (Barabási & Albert, 1999) graphs, which are formed by duplicating a BA graph multiple times and connecting copies of the same node in adjacent timesteps by an edge. The eigenvectors of such supra-Laplacian operators were recently used as positional encodings in temporal GNNs (Karmim et al., 2024; Galron et al., 2025). In these works, several eigenvectors (chosen between 16 and 128) were computed for every sampled time snapshot across the datasets during both training and inference. Examples of the two graph families are shown in Figure 4. The last graph family we use for the Laplacians is a synthetic "social hub" graph. The latter three topologies include some nodes that are substantially more connected than others, which often results in slow convergence or severe stencil growth. These patterns often arise

in power-law graphs, e.g., in social network analysis. Lastly, we train and test RAPNet on finite element discretizations of anisotropic and advection-diffusion PDEs in 3D.

Table 1 presents these results, showing that RAPNet provides a significant reduction in iteration counts in comparison to the classical baselines. Despite SpSA's theoretical robustness, we show that RAPNet achieves matching or superior convergence profiles. Moreover, RAPNet's GNN architecture inherently leverages GPU acceleration by design. We omit time measurements for solver iterations, since all three methods yield the same or equivalent sparsity patterns, and the iteration time is nearly identical for all methods. Figure 6 shows the convergence histories for a 3D advection-diffusion where RAPNet maintains rapid convergence while classical SpSA and AGG stall or fail.

Furthermore, AGG and SpSA use a sparse-LU exact coarsest-grid solver, while RAPNet uses Jacobi iterations on the coarsest grid to align with its training procedure. Despite using only approximate Jacobi iterations at the coarsest level, RAPNet remains competitive. See Section C for a detailed description of the graphs and operators used in our experiments and the parameters for their generation.

*Table 2.* Setup times in seconds averaged over 100 runs of 3D geometric, temporal Barabási-Albert, social hub and 3D advection-diffusion graphs. AGG and SpSA were run on a multi-core server-grade CPU, while RAPNet ran on an NVIDIA server-grade GPU. The time for RAPNet includes the initial time to run the aggregation algorithm (AGG), CPU-GPU data transfer time to transfer the AGG tensors, the execution of the GNN, and the on-GPU augmentation of the multigrid cycle using the emitted corrections.

|  | Vars | AGG (s) | SpSA (s) | RAPNet (s) |
|---|---|---|---|---|
| 3D Geo | 128K | 0.29 ± 0.01 | 0.73 ± 0.01 | 0.41 ± 0.01 |
|  | 256K | 0.59 ± 0.01 | 1.59 ± 0.02 | 0.85 ± 0.01 |
|  | 512K | 1.25 ± 0.04 | 3.50 ± 0.18 | 1.66 ± 0.07 |
| TBA | 200K | 0.38 ± 0.06 | 28.68 ± 1.96 | 0.57 ± 0.05 |
|  | 400K | 0.75 ± 0.04 | 57.01 ± 4.12 | 1.13 ± 0.03 |
|  | 600K | 1.11 ± 0.05 | 84.61 ± 6.08 | 1.67 ± 0.06 |
| SH | 100K | 0.16 ± 0.07 | 42.98 ± 14.50 | 0.23 ± 0.02 |
|  | 200K | 0.31 | 175.53 ± 68.00 | 0.47 |
|  | 300K | 0.53 ± 0.01 | 973.16 ± 70.00 | 0.76 ± 0.01 |
| 3D Adv-Diff | 227K | 0.54 ± 0.10 | 2.09 ± 0.40 | 0.82 ± 0.14 |
|  | 355K | 1.00 ± 0.15 | 3.77 ± 0.58 | 1.47 ± 0.22 |
|  | 524K | 1.47 ± 0.15 | 5.53 ± 0.58 | 2.02 ± 0.19 |

*Table 3.* Ablation study: impact of individual components on standalone solver performance for the 3D geometric graph Laplacian.

| Vars | RAPNet | w/o $\Delta P, \Delta R$ | w/o $\Delta A_c$ | w/o mix |
|---|---|---|---|---|
| 128K | **59 ± 5** | 99 ± 10 | 83 ± 7 | 68 ± 6 |
| 256K | **64 ± 6** | 108 ± 11 | 91 ± 9 | 74 ± 6 |
| 512K | **71 ± 6** | 113 ± 13 | 98 ± 11 | 81 ± 8 |

A core objective of our approach is to achieve scale-invariance, where the network is trained on relatively small systems and generalizes to significantly larger problems at inference time. Specifically, the network is trained on graphs consisting of 4000 nodes but evaluated on much larger domains. We employ multiple strategies: where possible, we utilize smaller-scale graphs generated using the same underlying processes; this occurs in the geometric and social hub datasets; for the PDE experiments, we train on small 2D discretizations and evaluate on large 3D discretizations of the same PDEs, achieving generalization across both size and dimension; Finally, the Watts-Strogatz and TBA graphs exhibit distinct structures at larger scales that do not occur at smaller scales. For this reason, in these experiments we extract random subgraphs from large-scale graphs for use in training and use a different set of large-scale graphs in evaluation. This exposes the network to the geometric and topological features it will encounter during evaluation while maintaining reasonable training dataset sizes. The procedure used to extract these subgraphs is detailed in Section B.2.

**Setup phase efficiency** Beyond iteration counts, we assess the practical overhead of the setup phase. As shown

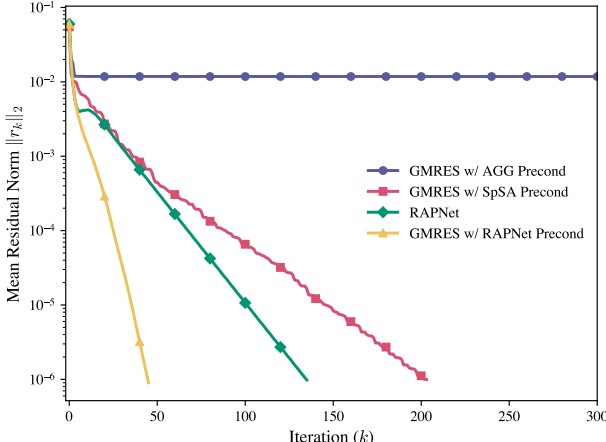

*Figure 6.* Convergence histories for an example 3D advection-diffusion system. RAPNet maintains rapid convergence while classical SpSA and AGG stall or fail.

in Table 2, applying the trained RAPNet introduces negligible latency over the baseline AGG method. Because our setup process starts by forming the aggregation (AGG), the marginal cost of RAPNet consists of a single GNN forward pass and sparse matrix additions for the corrections. Since the corrections share the same sparsity as the aggregation matrices, these operations are performed efficiently. Conversely, the results reveal a scalability challenge for SpSA on high-connectivity topologies. As an example, its setup time on the temporal Barabási-Albert and social hub datasets scales non-linearly. This is consistent with the algorithmic bottleneck discussed in Section 3.1.

**Ablation study** Lastly, we test the contribution of individual components to the performance of RAPNet. This study is shown in Table 3 for the 3D geometric dataset. The full network consistently achieves the fastest convergence, requiring fewer iterations compared to its variants.

# 6. Conclusion

We introduced RAPNet, a GNN framework for accelerating the solution of sparse linear systems. By learning sparse corrections to multigrid operators, RAPNet mitigates the trade-off between sparsity and convergence inherent to classical heuristics. RAPNet requires neural inference only in the setup phase without changing the solution process itself. In our experiments, RAPNet reduces the number of iterations for a diverse set of problems.

## Acknowledgments

This work was supported in part by the US–Israeli NSF-BSF award 2411264. Ido Ben-Yair is supported by the Kreitman High-Tech scholarship. Lars Ruthotto's work is supported in part by NSF award DMS-2038118. The authors also thank the Lynn and William Frankel Center for Computer Science at BGU.

## Impact Statement

This paper presents work whose goal is to advance the field of Machine Learning for computational mathematics applications. There are many potential societal consequences of our work, none of which we feel must be specifically highlighted here.

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

# A. Algebraic Multigrid

Algebraic multigrid uses coarser representations of the algebraic system $A$ to communicate information at different scales, achieving quick convergence as distant constraints in the system can be reconciled faster. From a deep learning perspective, AMG resembles a hierarchical graph neural network, where relaxations act as message-passing layers and transfer operators serve as graph pooling and unpooling (Eliasof & Treister, 2020). Any error vector decomposes into multiple frequency modes, generally divided into low- and high-frequency modes corresponding to small and large eigenvalues of the matrix, and their associated eigenvectors.

In a graph context, high-frequency error is highly oscillatory and local, i.e., varying rapidly between neighboring nodes, whereas low-frequency error is smooth and global. High-frequency error is easily handled by *smoothing*, also called *relaxation*, like the damped Jacobi method

$$\mathbf{x}^{(k+1)} = \mathbf{x}^{(k)} + \omega D^{-1} \left( \mathbf{b} - A\mathbf{x}^{(k)} \right), \qquad (14)$$

where $D$ is the diagonal of $A$ and $0 < \omega \leq 1$ is a damping parameter. Other iterative methods, such as Gauss-Seidel relaxation, are also commonly used. However, this smoothing fails to eliminate the smooth part of the error, because it typically corresponds to lower-energy eigenmodes of the discrete operator. To eliminate this low-frequency error, the residual is transferred to coarser scales where it appears as higher-frequency modes of the coarser system, enabling the re-application of smoothing at those scales.

Accordingly, starting from the finest level with system matrix $A_0 = A$, AMG recursively defines coarser levels $A_1, A_2, \ldots, A_L$ using the Galerkin product

$$A_{l+1} = R_l A_l P_l,$$

where $P_l$ is the *prolongation* (or *interpolation*) operator from level $l + 1$ to level $l$, and $R_l$ is the *restriction* operator from level $l$ to level $l + 1$. This standard Galerkin formulation has an important variational interpretation. When the system matrix $A_l$ is symmetric positive-definite (SPD) and the restriction is chosen as the transpose of the prolongation, i.e., $R_l = P_l^T$, solving the coarse-grid problem corresponds to an orthogonal projection of the exact error. Specifically, it ensures that the coarse-grid correction minimizes the $A$-norm (or energy norm) of the error, $\|\mathbf{e}_l\|_{A_l} = \sqrt{\mathbf{e}_l^T A_l \mathbf{e}_l}$, over all possible corrections in the subspace defined by $P_l$.

This guarantees that we find the optimal linear combination of the coarse basis vectors to reduce the current residual. The choice of $P_l$ and $R_l$ is the main differentiator between various multigrid methods, and their quality directly impacts the convergence rate of the solver. Thus, designing effective transfer operators is a central challenge in AMG research. See Briggs et al. (2000) for a more detailed discussion.

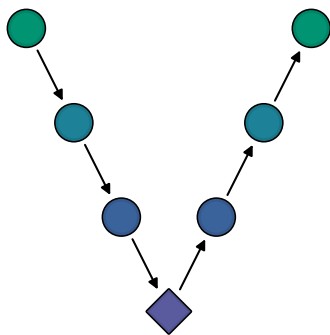

*Figure 7.* **AMG V-Cycle.** The V-cycle operates recursively starting from the finest level (green circles), culminating in the direct solution of the coarsest system at the coarse level (purple diamond).

Given a right-hand side vector $\mathbf{b}$, AMG solves the problem $A\mathbf{x} = \mathbf{b}$ using recursive approaches, the simplest of which is the *V-cycle*. The V-cycle begins from the finest level $l = 0$, and applies a few iterations of the chosen smoother to the initial solution $\mathbf{x}_l$ (at level $l$) to reduce high-frequency error components, resulting in an improved $\mathbf{x}_l$. This step, called the *pre-relaxation* or *pre-smoothing* step, is often applied as in (14).

Next, the algorithm derives the correction to reduce the remaining residual $\mathbf{r}_l = \mathbf{b} - A_l \mathbf{x}_l$. It achieves this by computing a solution to the coarse error-residual equation

$$A_{l+1} \mathbf{e}_{l+1} = \mathbf{r}_{l+1},$$

where $\mathbf{r}_{l+1}$ is obtained by restricting $\mathbf{r}_l$ to the coarse level using $R_l$, i.e., $\mathbf{r}_{l+1} = R_l \mathbf{r}_l$. This projected error-residual equation, also termed the *coarse-grid problem*, is solved either directly if the system is small enough, or by a recursive application of the multigrid algorithm. The coarse system solution can also be approximated by other methods.

After solving the coarse-level system, the solution $\mathbf{e}_{l+1}$ is interpolated back to the finer level with the prolongation $P_l$, i.e., $\mathbf{e}_l = P_l \mathbf{e}_{l+1}$ and added to the current approximation of the solution, i.e., $\mathbf{x}_l + P_l \mathbf{e}_{l+1}$. Finally, a few more smoothing iterations are applied, known as the *post-relaxation* or *post-smoothing* step.

AMG essentially applies two complementary operations at each level: smoothing to reduce high-frequency error and a *coarse-grid correction* to address low-frequency error. This requires that the correction for the smooth error modes is approximately in the range of the interpolation, i.e., $P_l \mathbf{e}_{l+1} \approx \mathbf{e}_l$ for some smooth $\mathbf{e}_l$. Since this is often not the case, some error components may not be fully eliminated. Furthermore, because the interpolation usually "pollutes" the solution with high-frequency error, the post-smoothing step may be necessary to further correct the solution.

The two-level error iteration matrix of the AMG V-cycle is given succinctly by

$$\mathbf{e}^{(k+1)} = S^{\nu_2}(I - P_l A_{l+1}^{-1} R_l A_l)S^{\nu_1}\mathbf{e}^{(k)}, \quad (15)$$

where $\mathbf{e}^{(k)}$ is the error at the $k$-th iteration, $S^{\nu_1}$ is the pre-smoother applied $\nu_1$ times, and $S^{\nu_2}$ denotes the post-smoother applied $\nu_2$ times. Although the pre- and post-smoothers can be distinct, we have denoted both of them by $S$ here for simplicity. When applying more than two levels, the coarse-grid solve $A_{l+1}^{-1}$ is replaced by a recursive application of (15). The complete algebraic multigrid V-cycle algorithm is given in Algorithm 1 and its structure is visualized in Figure 7.

The structure of AMG resembles a hierarchical GNN, i.e., a graph U-Net. This natural similarity enables the combination of AMG and GNN architectures here and in many of the cited works.

---

**Algorithm 1** Algebraic Multigrid V-Cycle

**Input:** Matrix $A_l$, RHS $\mathbf{b}_l$, init. sol. $\mathbf{x}_l$, level $l$
**if** $l$ is the coarsest level **then**
   Solve $A_l\mathbf{x}_l = \mathbf{b}_l$ directly
**else**
   $\mathbf{x}_l \leftarrow$ Pre-smooth$(A_l, \mathbf{b}_l, \mathbf{x}_l)$
   $\mathbf{r}_{l+1} \leftarrow R_l(\mathbf{b}_l - A_l\mathbf{x}_l)$
   $\mathbf{e}_{l+1} \leftarrow$ V-Cycle$(A_{l+1}, \mathbf{r}_{l+1}, \mathbf{0}, l+1)$
   $\mathbf{x}_l \leftarrow \mathbf{x}_l + P_l\mathbf{e}_{l+1}$
   $\mathbf{x}_l \leftarrow$ Post-smooth$(A_l, \mathbf{b}_l, \mathbf{x}_l)$
**end if**
**Output:** Updated solution $\mathbf{x}_l$

---

## B. Architecture, Training and Implementation

We provide specific details and hyperparameter configurations used to implement, train, and evaluate the RAPNet models discussed in Section 5. We implemented RAPNet using PyTorch Geometric (Fey & Lenssen, 2019), PyTorch (Paszke et al., 2019) and SciPy sparse matrix infrastructure (Virtanen et al., 2020). We support our experiments with our novel implementation of smoothed aggregation and SpSA. The latter was ported from the original MATLAB/MEX code. We used the RGGCN implementation from the GraphGPS paper (Rampášek et al., 2022). FEM systems were created using DOLFINx (Baratta et al., 2023) and graphs were generated using NetworkX (Hagberg et al., 2008). Visualizations were created with matplotlib (Hunter, 2007), NetworkX and draw.io (JGraph, 2021).

Training each of the seven models (one model per dataset type, each with a parameter count of ∼110K) required between 3 and 6 hours. We performed model selection based on performance on a held-out validation set across all training epochs. We used NVIDIA server-grade GPUs with

*Table 4.* Common training hyperparameters

| Parameter | Value |
|---|---|
| Optimizer | SGD/Adam |
| Learning Rate | $1 \times 10^{-4}$ |
| Weight Decay | 0 |
| Epochs | 30 |
| Batch Size (Operators) | 8 |
| Batch Size (RHS) | 64 |
| Precision | Single |

*Table 5.* Structural parameters of the RAPNet architecture. All internal MLP hidden layers and RGGCN embeddings maintain a constant width of 64 channels.

| Component | Parameter | Value |
|---|---|---|
| General | Hidden Dimension | 64 |
| | Activation Function | ReLU |
| Encoders | Node MLP Layers | 3 |
| | Edge MLP Layers | 3 |
| Processor | RGGCN Layers | 3 |
| | Residual Connections | Enabled |
| Decoder | Edge MLP Layers | 3 |
| Mixing | Node Linear | $128 \rightarrow 64$ |
| | Edge Linear | $128 \rightarrow 64$ |

48GB of VRAM for both training and inference, supported by server-grade multi-core CPUs.

### B.1. Common Hyperparameters

Across all experiments, we maintained a consistent relaxation configuration: Jacobi smoothing with damping factor of $\omega = 0.6$. The SA filtering parameters remained fixed at $\epsilon_{\mathrm{soc}} = 0.5$ (strength-of-connection filtering threshold) and $\epsilon_{\mathrm{mat}} = 0.02$ (matrix filtering threshold). Table 4 summarizes the shared hyperparameters used across all experiments.

The complete structural configurations, including layer depths and channel dimensions for the encoders, processor, and decoder, are summarized in Table 5.

### B.2. Subgraph Extraction Procedure

To train our models on self-similar localized structures within large-scale domains, we utilize a randomized sampling procedure based on breadth-first search (BFS) starting from a random node. Given a global graph $\mathcal{G} = (\mathcal{V}, \mathcal{E})$ and a target node budget $k$, the procedure extracts an induced subgraph $\mathcal{G}_{\mathrm{sub}}$ that preserves local connectivity and structural properties. A naive BFS without these modifications produced training subgraphs that were too topologically uniform or lacked crucial featuers, leading to poor generalization. The process described here ensures that the sampled

*Table 6.* Summary of experiment-specific training parameters.

| Dataset | Samples | Nodes | Norm. Ops |
|---|---|---|---|
| 2D Geometric | 400 | 4000 | No |
| 3D Geometric | 800 | 4000 | No |
| Watts-Strogatz | $32 \times 32$ | 4000 | No |
| TBA | $400 \times 4$ | 4000 | No |
| Social Hub | 400 | 4000 | No |
| 3D Aniso Diff | 800 | $64 \times 64$ | Yes |
| 3D Adv-Diff | 800 | $64 \times 64$ | Yes |

*Table 7.* AMG hierarchy depth and V-cycle smoother iteration configurations per experiment. $L_{\text{train}}$ and $L_{\text{eval}}$ denote the hierarchy depth during training and inference. $S_{\text{train}}$ and $S_{\text{eval}}$ denote the smoother iterations used during training and evaluation.

| Dataset | $L_{\text{train}}$ | $L_{\text{eval}}$ | $S_{\text{train}}$ | $S_{\text{eval}}$ |
|---|---|---|---|---|
| 2D Geometric | 4 | 5 | 2 | 1 |
| 3D Geometric | 4 | 5 | 2 | 1 |
| Watts-Strogatz | 5 | 6 | 2 | 1 |
| TBA | 4 | 5 | 2 | 1 |
| Social Hub | 4 | 4 | 2 | 2 |
| 3D Aniso Diff | 4 | 5 | 2 | 1 |
| 3D Adv-Diff | 4 | 5 | 2 | 2 |

nodes form a spatially cohesive cluster, prevents missing salient graph structures, and mitigates needless repetition in the final dataset.

The key features of this process are:

- **Direction-agnostic traversal:** If $\mathcal{G}$ is directed, i.e., arising from an asymmetric matrix, we perform the traversal on an undirected view $\mathcal{G}_{\text{undirected}}$ where an edge exists between $u$ and $v$ if $(u, v) \in \mathcal{E}$ or $(v, u) \in \mathcal{E}$. This prevents the BFS from being "trapped" in leaf or sink nodes.

- **Stochastic neighbor selection:** To avoid biases resulting from the original node ordering, the neighbors of the current node $v$ are randomly shuffled before being appended to the BFS queue.

- **Handling disjoint components:** If the traversal has exhausted a connected component before the target budget $k$ is reached, a new root node is selected uniformly at random from the remaining unvisited set $\mathcal{V} \setminus \mathcal{V}_{\text{visited}}$.

- **Subgraph induction:** Once $|\mathcal{V}_{\text{visited}}| = k$, we take $\mathcal{G}_{\text{sub}}$ as the node-induced subgraph of the original graph $\mathcal{G}$. This ensures that all original edge attributes and directions are preserved for the final matrix $A_{\text{sub}}$.

## C. Dataset-Specific Details

We evaluated our model across seven distinct datasets, spanning geometric graphs, complex network topologies, and physical PDE systems. Table 6 provides a comparative summary of the training parameters for each experiment.

During inference, we evaluate the performance of RAP-Net using deeper V-cycles than those encountered during the training phase, and using a different number of Jacobi smoother iterations. The specific training and evaluation depths, along with the number of smoother iterations for each experiment, are detailed in Table 7.

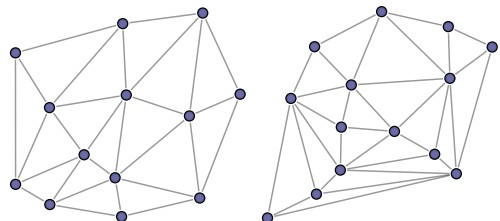

*Figure 8.* **2D geometric graph examples.**

### C.1. Geometric Graphs (2D & 3D)

The geometric datasets consist of random point meshes generated in $d$-dimensional Euclidean space ($d \in \{2, 3\}$). For each sample, 4000 points were sampled from a uniform distribution within a unit hypercube $[0, 1]^d$. To ensure well-defined boundaries, bounding points were added at the vertices of the unit square or cube. The graph adjacency structure was established via Delaunay triangulation or tetrahedralization of the sampled points. This approach ensures a planar-like graph structure in 2D and a tetrahedral mesh structure in 3D, providing a realistic proxy for discretized physical domains. Examples of these graphs are shown in Figure 8.

### C.2. Watts-Strogatz (Small-World)

Watts-Strogatz graphs exhibit small-world properties, characterized by high local clustering and short average path lengths, which are controlled by a rewiring probability $p$. This combines a regular lattice structure with random long-range connections, hence "small-world". We generated 32 large graphs ($n = 512,000, p = 0.01, k = 6$) and sampled 32 subgraphs of 4000 nodes from each.

### C.3. Temporal Barabási-Albert (Scale-Free)

These graphs represent scale-free networks generated via preferential attachment. The dataset was constructed by first generating 400 base Barabási-Albert graphs ($n = 4000, m = 2$). To incorporate temporal dynamics, each graph was extended over 50 discrete time steps by construct-

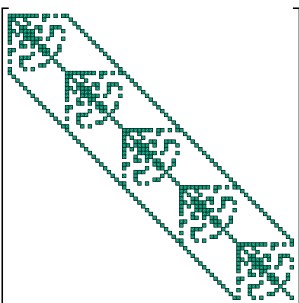

*Figure 9.* **Supra-Laplacian sparsity.** The block-diagonal structure represents the intra-layer topology of the Barabási-Albert graphs across 5 time steps, while the off-diagonal bands indicate the coupling between consecutive temporal layers.

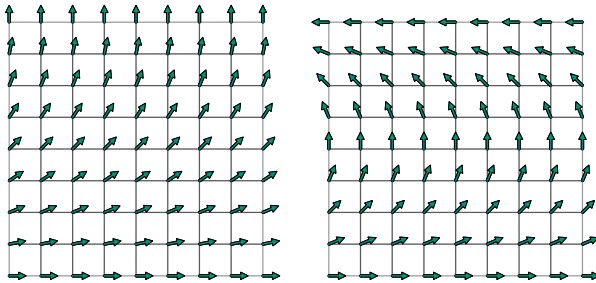

*Figure 10.* **Training set vector field visualizations for the PDE datasets. (Left)** A velocity field for advection-diffusion. **(Right)** An orientation field for anisotropic diffusion.

ing a supra-Laplacian operator, which encodes both the intra-layer topology and inter-layer temporal dependencies. From each of these 400 extended systems, we sampled 4 subgraphs of 4000 nodes each, resulting in a diverse training set of 1600 samples. An example sparsity pattern of this operator is illustrated in Figure 9.

### C.4. Synthetic Social Hub

These graphs model typical social network topologies by introducing highly-connected social hubs into a random geometric base. Specifically, we extended the random 2D geometric graphs by adding $h = 5$ hub nodes. Each hub is connected to $65\%$ of a targeted sub-population that comprises half of the total nodes. This structure facilitates rapid signal propagation and significantly reduces the overall diameter of the graph. Because these graphs require a small number of iterations to converge due to their high connectivity, we use 100 iterations of Jacobi for the coarse grid solver instead of the usual 2.

### C.5. Anisotropic Diffusion and Advection-Diffusion

#### C.5.1. TRAINING

These PDE experiments utilized structured $64 \times 64$ grids for training, with the operators normalized. We found experimentally that this normalization was vital for stability in these problems, where the operator can become highly anisotropic, i.e., non-symmetric. Both problems included varying directions—anisotropy for diffusion and flow for advection—to ensure the model remains robust to directional bias. Examples of velocity fields for anisotropic diffusion and advection-diffusion problems used in training are shown in Figure 10.

**Anisotropic Diffusion**   We train the model using data generated from the anisotropic diffusion equation with Neu-

mann boundary conditions on the unit square $\Omega = [0, 1]^2$:

$$-\nabla \cdot (\sigma(x, y)\nabla u) = f,$$

where the diffusion tensor $\sigma$ is defined as

$$\sigma(x, y) = R(\theta(y)) \begin{bmatrix} 1 & 0 \\ 0 & 10^{-4} \end{bmatrix} R(\theta(y))^T,$$

and the rotation matrix $R(\theta)$ is given by

$$R(\theta) = \begin{bmatrix} \cos\theta & -\sin\theta \\ \sin\theta & \cos\theta \end{bmatrix}.$$

The rotation angle is given by $\theta(y) = (\theta_0 + (\theta_1 - \theta_0)y)\pi$, where $\theta_0$ is sampled uniformly from $[0, 2)$ and $\theta_1 = \theta_0 + \delta$, with $\delta \in [-8/9, 8/9]$.

**Advection-Diffusion**   We train the model using data generated from the steady-state advection-diffusion equation with Dirichlet boundary conditions on the unit square $\Omega = [0, 1]^2$:

$$-\varepsilon\nabla^2 u + \mathbf{v} \cdot \nabla u = f,$$

where the diffusion coefficient is $\varepsilon = 10^{-4}$. Training is performed with the velocity field

$$\mathbf{v}(x, y) = \begin{bmatrix} \cos\big((\theta_0 + (\theta_1 - \theta_0)y)\pi\big) \\ \sin\big((\theta_0 + (\theta_1 - \theta_0)y)\pi\big) \end{bmatrix},$$

where $\theta_0$ is sampled uniformly from $[0, 2)$ and $\theta_1 = \theta_0 + \delta$, with $\delta \in [-1/6, 1/6]$.

#### C.5.2. EVALUATION

For the PDE experiments, training was done on 2D structured meshes while evaluation was performed on real-world, unstructured, 3D meshes. The base domain is a unit cube $\Omega = [0, 1]^3$, which is subsequently modified by applying cutouts. The dataset comprises three primary categories of spatial configurations with varying degrees of topological complexity and internal boundaries. All domains are spatially discretized into 3D unstructured tetrahedral meshes.

The spatial resolution of the mesh is bounded by a specified minimum and maximum element size of 0.008-0.01 and 0.1-0.5, respectively. Gmsh was used to create these unstructured meshes.

The generation process is as follows:

- **Baseline and Edge Cases (Domains 1 and 2):** The first domain is an unmodified, solid unit cube with no internal cutouts, serving as the trivial baseline. The second domain features a complex arrangement of five uniform spherical cutouts positioned at specific coordinates $p$ to test flow/diffusion around clustered obstacles where

$$p = (0.25, 0.25, 0.25),$$
$$p = (0.75, 0.25, 0.25),$$
$$p = (0.5, 0.5, 0.5),$$
$$p = (0.25, 0.75, 0.75), \text{ or}$$
$$p = (0.75, 0.75, 0.75).$$

- **Rotated Cuboid Cutouts:** To evaluate the model's response to sharp corners and varied boundary orientations, we introduce rotated box cutouts. These geometries are sampled from a combinatorial space parameterized by their center coordinates

$$c_x, c_y, c_z \in \{0.0, 0.25, 0.35, 0.5, 0.75\},$$

their side lengths $h \in \{0.35, 0.5\}$ and their rotation angles

$$\theta_x, \theta_y, \theta_z \in \{°, 15°, 30°, 45°\}.$$

These account for approximately 50% of remaining domains.

- **Spherical Cutouts:** To evaluate the model's response to smooth boundaries of varying scales, we introduce single spherical cutouts where the center coordinates are

$$c_x, c_y, c_z \in \{0.0, 0.25, 0.35, 0.5, 0.75\}$$

and the radii are

$$r \in \{0.15, 0.2, 0.25, 0.35, 0.45\}.$$

These account for approximately 50% of remaining domains.

With this procedure, we sample 100 examples. To guarantee a diverse subset of geometries, the parameter arrays (offsets, heights, angles, and radii) are randomly shuffled prior to computing their Cartesian product. This ensures that the generated subset avoids structural bias (e.g., generating only small cutouts or cutouts clustered in one corner of the domain) and exhibits high geometric variance across the final meshed domains.

**Anisotropic Diffusion**   We solve the 3D anisotropic diffusion equation with Dirichlet boundary conditions on the unit cube $\Omega = [0, 1]^3$:

$$-\nabla \cdot (\sigma(x, y, z)\nabla u) = f,$$

where the diffusion tensor $\sigma$ is

$$\sigma(x, y, z) = R(\theta, \phi) \begin{bmatrix} 1 & 0 & 0 \\ 0 & 10^{-4} & 0 \\ 0 & 0 & 10^{-4} \end{bmatrix} R(\theta, \phi)^T.$$

The 3D rotation matrix $R(\theta, \phi) = R_z(\theta)R_y(\phi)$ is constructed from the composition of rotations along the Z and Y axes:

$$R_z(\theta) = \begin{bmatrix} \cos\theta & -\sin\theta & 0 \\ \sin\theta & \cos\theta & 0 \\ 0 & 0 & 1 \end{bmatrix}$$

and

$$R_y(\phi) = \begin{bmatrix} \cos\phi & 0 & -\sin\phi \\ 0 & 1 & 0 \\ \sin\phi & 0 & \cos\phi \end{bmatrix}.$$

The rotation angles $\theta(x, y, z)$ and $\phi(x, y, z)$ are chosen to align the primary diffusion axis with a prescribed velocity field $\mathbf{v}(x, y, z) = [v_x, v_y, v_z]^T$, computed as

$$\theta = \arctan2(v_y, v_x), \quad \phi = \arctan2\left(v_z, \sqrt{v_x^2 + v_y^2}\right),$$

where the underlying field is

$$\mathbf{v}(x, y, z) = \begin{bmatrix} x(1-y)(2-z) \\ y(1-z)(2-x) \\ z(1-x)(2-y) \end{bmatrix}.$$

**Advection-Diffusion**   We solve the steady-state advection-diffusion equation with Dirichlet boundary conditions on the unit cube $\Omega = [0, 1]^3$:

$$-\varepsilon\nabla^2 u + \mathbf{v} \cdot \nabla u = f,$$

where the diffusion coefficient is $\varepsilon = 10^{-4}$, and the 3D velocity field is

$$\mathbf{v}(x, y, z) = \begin{bmatrix} x(1-2y)(1-z) \\ y(1-2z)(1-x) \\ z(1-2x)(1-y) \end{bmatrix}.$$

## D. Supplementary Numerical Experiments

While RAPNet is not intended to solve these problems in particular (geometric multigrid or classical AMG are often preferred), we test RAPNet on linear elasticity systems. We use V-cycle hierarchies generated by the network trained on anisotropic diffusion problems and show that they can be successfully applied to the linear elasticity saddle-point

*Table 8.* Number of iterations to convergence of 2D elasticity systems where $\lambda = 10, \mu = 1$, standalone and under GMRES.

| Vars | AGG | SpSA | RAPNet |
|---|---|---|---|
| Standalone | | | |
| $128^2$ | 202 | Diverges | **139** |
| $256^2$ | 421 | Diverges | **245** |
| $512^2$ | 425 | Diverges | **292** |
| $1024^2$ | 681 | Diverges | **384** |
| GMRES | | | |
| $128^2$ | 68 | **50** | 55 |
| $256^2$ | 128 | **65** | 102 |
| $512^2$ | 135 | **68** | 112 |
| $1024^2$ | 212 | **83** | 155 |

*Table 9.* Number of iterations to convergence of 2D Stokes systems where $\lambda = 10^8, \mu = 1$, standalone and under GMRES.

| Vars | AGG | SpSA | RAPNet |
|---|---|---|---|
| Standalone | | | |
| $128^2$ | 393 | Diverges | **365** |
| $256^2$ | 394 | Diverges | **367** |
| $512^2$ | 374 | Diverges | **335** |
| $1024^2$ | 371 | Diverges | **330** |
| GMRES | | | |
| $128^2$ | **98** | Stalls | 103 |
| $256^2$ | 106 | Stalls | **104** |
| $512^2$ | **107** | Stalls | 138 |
| $1024^2$ | **104** | Stalls | 113 |

formulation as well. We approximate the inverse of the principal submatrices by applying one V-cycle. In each iteration, we apply one V-cycle for each scalar unknown, i.e., 3 V-cycles in 2D. For linear elasticity, all principal submatrices are standard discrete Laplacians.

The experiments for the linear elasticity saddle-point formulation and Stokes equation (i.e., where $\lambda \gg \mu$) are given in Tables 8 and 9.

The linear elasticity equation in an isotropic medium is given by

$$\nabla \lambda \nabla \cdot \mathbf{u} + \vec{\nabla} \cdot \mu \left( \vec{\nabla}\mathbf{u} + \vec{\nabla}\mathbf{u}^T \right) = \mathbf{q},$$

where $\mathbf{u} = \mathbf{u}(\mathbf{x})$ is the displacement vector.

An alternative formulation is

$$\vec{\nabla} \cdot \mu \vec{\nabla}\mathbf{u} + \nabla(\lambda + \mu)\nabla \cdot \mathbf{u} = \mathbf{q},$$

obtained by applying the equality

$$\nabla \cdot \left( \vec{\nabla}\mathbf{u} + \vec{\nabla}\mathbf{u}^T \right) = \vec{\nabla} \cdot \vec{\nabla}\mathbf{u} + \nabla\nabla \cdot \mathbf{u},$$

which holds in continuous space. For some popular discretizations, this equality also holds in discrete space (e.g., MAC discretization on staggered grids).

The two formulations are equivalent in homogeneous media, and resemble each other in heterogeneous media. The coefficients $\lambda$ and $\mu$ are the Lamé coefficients, which represent the stress-strain relationship in the material.

The elasticity equation yields a system of equations which has a rich near-nullspace due to the presence of the $\nabla(\lambda + \mu)\nabla\cdot$ operator. If $\lambda \gg \mu$, i.e., this operator is dominant, then iterative methods struggle. Fortunately, by introducing a pressure variable $p = -(\lambda + \mu)\nabla \cdot u$, the system can be transformed into the following "regularized" *saddle-point system*, also called a *mixed formulation* (Gaspar et al., 2008). For a constant $\mu$, we get

$$\begin{bmatrix} -\mu\vec{\Delta} & \nabla \\ -\nabla\cdot & -\frac{1}{\lambda+\mu} \end{bmatrix} \begin{bmatrix} \mathbf{u} \\ p \end{bmatrix} = \begin{bmatrix} -\mathbf{q} \\ 0 \end{bmatrix}.$$

This transformation enables us to apply recent methods from fluid dynamics research. One solution is to use block-wise (e.g., cell-wise) relaxation within multigrid, also known as *Vanka relaxation*. This method inverts each local submatrix for each block (cell) instead of applying the pointwise update in Jacobi. While robust, it is considered expensive, since it requires inverting matrices for each cell. For example, in 3D these matrices will be $7 \times 7$.

*Block preconditioning* is often a more effective solution. Its purpose is to transform the original coupled system into a block-triangular system, which is then solved for each scalar variable separately, i.e., by applying the commutator equality

$$\nabla \cdot \vec{\Delta} = \Delta\nabla\cdot, \tag{16}$$

which also holds in continuous space for constant coefficients. However, for heterogeneous coefficients, it is only approximate.

Taking the divergence of the first equation yields

$$-\nabla \cdot \mu\vec{\Delta}\mathbf{u} + \nabla \cdot \nabla p = -\nabla \cdot \mathbf{q}$$

and then applying the commutator equality and definition of $p$ yields

$$-\mu\Delta\nabla \cdot \mathbf{u} + \nabla \cdot \nabla p = -\nabla \cdot \mathbf{q},$$

which equals

$$-\frac{\lambda + 2\mu}{\lambda + \mu}\Delta p = -\nabla \cdot \mathbf{q}. \tag{17}$$

Equation (17) is a Poisson equation for $p$, often known as the *pressure Poisson* equation in fluid dynamics. We can apply the same technique to form a preconditioner in discrete space where the above equalities are approximate. This is possible because the equalities above are accurate in continuous space and form an exact elimination in that space. In a discrete space, they form an approximate elimination.

To form a preconditioner, we follow the derivation of (Yovel & Treister, 2026). Let us denote the discrete error-residual version of the system above as

$$\begin{bmatrix} A & B^T \\ B & -C \end{bmatrix} \begin{bmatrix} \mathbf{e}_u \\ e_p \end{bmatrix} = \begin{bmatrix} \mathbf{r}_u \\ r_p \end{bmatrix}. \quad (18)$$

The left-preconditioned system is then

$$\begin{bmatrix} I & 0 \\ B & -A_p \end{bmatrix} \begin{bmatrix} A & B^T \\ B & -C \end{bmatrix} \begin{bmatrix} \mathbf{e}_u \\ e_p \end{bmatrix} = \begin{bmatrix} I & 0 \\ B & -A_p \end{bmatrix} \begin{bmatrix} \mathbf{r}_u \\ r_p \end{bmatrix},$$

where $-A_p = \mu BB^T = \mu \Delta_h$ is the discrete Laplacian operator for the $p$ variable.

Using the commutativity property in (16), $BA = A_p B$ and hence the equation above becomes

$$\begin{bmatrix} A & B^T \\ 0 & BB^T + A_p C \end{bmatrix} \begin{bmatrix} \mathbf{e}_u \\ e_p \end{bmatrix} = \begin{bmatrix} I & 0 \\ B & -A_p \end{bmatrix} \begin{bmatrix} \mathbf{r}_u \\ r_p \end{bmatrix}.$$

The bottom equation here is the same as in (17), only for a different right-hand side, coming from the error-residual equation (18). The bottom-left zero block is only approximate, and hence the solution will take a few iterations, even if the blocks are inverted exactly.

## E. Supplementary Computational Analysis

RAPNet builds on the sparsity pattern computed by AGG. Thus, their operator complexities are identical. In contrast, the sparsity patterns produced by SpSA are subject to small variations. This occurs because SpSA introduces slight modifications to edge weights during hierarchy construction, leading to edge rewiring that can alter the strength-of-connection metric at subsequent levels. The relative operator complexities for the largest experiments are given in Table 10.

In the main text, the setup times for RAPNet are reported with GPU acceleration, reflecting its optimal deployment in modern machine learning workflows. However, traditional HPC environments often do not include GPU hardware, instead relying on a large number of interconnected CPUs. In these cases, RAPNet inference can be quickly executed on CPUs, since it is done only once at the initial setup and the network consists only of roughly 110K learnable parameters. Example setup times for this scenario, evaluated on a consumer-grade CPU, are shown in Table 11.

Finally, Table 12 details the overall convergence rates across the evaluated datasets. As shown, RAPNet converges reliably. These results confirm that the accelerated iteration counts do not come at the cost of overall solver stability.

Table 10. Average operator complexities for all solvers, averaged over 100 examples.

|  | Vars | NNZ | AGG&RN | SpSA |
|---|---|---|---|---|
| 2D Geometric | 128K | 896K | 1.12 | 1.11 |
|  | 256K | 1.8M | 1.12 | 1.11 |
|  | 512K | 3.6M | 1.12 | 1.12 |
|  | 1M | 7M | 1.13 | 1.12 |
| 3D Geometric | 128K | 2.1M | 1.05 | 1.05 |
|  | 256K | 4.2M | 1.05 | 1.05 |
|  | 512K | 8.4M | 1.05 | 1.05 |
| Watts-Strogatz | 128K | 896K | 1.14 | 1.13 |
|  | 256K | 1.8M | 1.15 | 1.13 |
|  | 512K | 3.6M | 1.16 | 1.13 |
|  | 1M | 7M | 1.18 | 1.13 |
| Temporal Barabási-Albert | 200K | 1.4M | 1.71 | 1.65 |
|  | 400K | 2.8M | 1.71 | 1.65 |
|  | 600K | 4.2M | 1.72 | 1.65 |
| Social Hub | 100K | 1M | 1.49 | 4.34 |
|  | 200K | 2.1M | 1.85 | 8.91 |
|  | 300K | 3.1M | 2.21 | 13.80 |
| 3D FEM Anisotropic Diffusion | 227K | 2.7M | 1.37 | 1.37 |
|  | 355K | 4.3M | 1.35 | 1.34 |
|  | 524K | 6.5M | 1.34 | 1.34 |
| 3D FEM Advection-Diffusion | 227K | 2.7M | 1.31 | 1.31 |
|  | 355K | 4.3M | 1.28 | 1.29 |
|  | 524K | 6.5M | 1.27 | 1.28 |

Table 11. Average CPU setup times for RAPNet, averaged over 100 examples.

|  | Vars | Setup Time (s) |
|---|---|---|
| 3D Geometric | 128K | 12.94 ± 0.18 |
|  | 256K | 26.14 ± 0.30 |
|  | 512K | 62.72 ± 2.51 |
| Temporal Barabási-Albert | 200K | 23.07 ± 0.38 |
|  | 400K | 46.86 ± 0.51 |
|  | 600K | 69.93 ± 1.00 |
| Social Hub | 100K | 8.81 ± 0.12 |
|  | 200K | 18.01 ± 0.42 |
|  | 300K | 26.96 ± 0.18 |
| 3D FEM Advection-Diffusion | 227K | 25.22 ± 4.10 |
|  | 355K | 43.33 ± 5.54 |
|  | 524K | 62.41 ± 5.57 |

*Table 12.* Convergence rates of the solvers across all evaluated datasets.

| | Vars | NNZ | Standalone | | | GMRES | | |
| --- | --- | --- | --- | --- | --- | --- | --- | --- |
| | | | AGG | SpSA | RAPNet | AGG | SpSA | RAPNet |
| 2D Geometric | 128K | 896K | 100% | 100% | 100% | 100% | 100% | 100% |
| | 256K | 1.8M | 100% | 96% | 100% | 100% | 100% | 100% |
| | 512K | 3.6M | 100% | 99% | 100% | 100% | 100% | 100% |
| | 1M | 7M | 100% | 97% | 100% | 100% | 100% | 100% |
| 3D Geometric | 128K | 2.1M | 100% | 100% | 100% | 100% | 100% | 100% |
| | 256K | 4.2M | 100% | 100% | 100% | 100% | 100% | 100% |
| | 512K | 8.4M | 100% | 100% | 100% | 100% | 100% | 100% |
| Watts-Strogatz | 128K | 896K | 99% | 61% | 99% | 69% | 72% | 100% |
| | 256K | 1.8M | 98% | 57% | 100% | 57% | 63% | 100% |
| | 512K | 3.6M | 100% | 40% | 100% | 56% | 65% | 100% |
| | 1M | 7M | 100% | 18% | 100% | 54% | 51% | 100% |
| Temporal | 200K | 1.4M | 100% | 100% | 100% | 100% | 100% | 100% |
| Barabási-Albert | 400K | 2.8M | 100% | 100% | 100% | 100% | 100% | 100% |
| | 600K | 4.2M | 100% | 100% | 100% | 100% | 100% | 100% |
| Social Hub | 100K | 1M | 100% | 51% | 100% | 100% | 100% | 100% |
| | 200K | 2.1M | 100% | 48% | 100% | 100% | 100% | 100% |
| | 300K | 3.1M | 100% | 6% | 100% | 100% | 100% | 100% |
| 3D FEM | 227K | 2.7M | 100% | 100% | 100% | 100% | 100% | 100% |
| Anisotropic | 355K | 4.3M | 100% | 100% | 100% | 100% | 100% | 100% |
| Diffusion | 524K | 6.5M | 100% | 100% | 100% | 100% | 100% | 100% |
| 3D FEM | 227K | 2.7M | 0% | 3% | 100% | 100% | 100% | 100% |
| Advection- | 355K | 4.3M | 14% | 22% | 100% | 99% | 100% | 100% |
| Diffusion | 524K | 6.5M | 10% | 13% | 100% | 100% | 100% | 100% |

