# OpenReview forum: "RAPNet: Accelerating Algebraic Multigrid with Learned Sparse Corrections"
_ICML.cc/2026/Conference — ICML 2026 regular_

### Official Review · Reviewer_QZQC · 2026-03-02

**Soundness:** 2
**Presentation:** 4
**Significance:** 2
**Originality:** 4
**Overall Recommendation:** 3
**Confidence:** 5

**Summary:**

This paper proposes RAPNet, a GNN-based framework that learns to generate sparse and robust coarse operators directly from a sparse algebraic system. The goal is to balance sparsity with coarse-grid quality (i.e., convergence) when constructing coarse operators. RAPNet is trained on small local patches but is claimed to generalize to million-node systems; at inference it runs once per system and can be reused for multiple right-hand sides. The method is evaluated on a range of problem types, both as a standalone solver component and as a preconditioner.

**Compliance With Llm Reviewing Policy:**

Affirmed.

**Final Justification:**

I recommend **Weak Reject (3)**. RAPNet is an original and practically motivated idea, using a GNN to learn **sparse coarse operators** from the algebraic system which is paired with a solid experimental suite at meaningful scales (including million-node settings and multiple RHS reuse). The paper is also clearly written with good AMG background and visualizations, which supports **clarity** and makes the contribution easy to follow.

However, I weigh the weaknesses as more important for acceptance. First, RAPNet is still tightly coupled to the **classical AMG pipeline** (unsmoothed aggregation to initialize the hierarchy), so it is not fully clear how much improvement comes from learning versus inheriting the strongest classical ingredients. Second, in the most common AMG deployment, **preconditioning GMRES**, RAPNet is often weaker than SpSA. The rebuttal clarifies the intended framing (setup-time augmentation; argue for time-to-solution/memory), but does not provide a sufficiently **systematic end-to-end** comparison (setup+solve time, peak memory across regimes) to establish clear practical superiority. As a result, **soundness/significance** remain only fair despite good originality and presentation.

**Key Questions For Authors:**

1. Since AMG is most commonly used as a preconditioner, why does RAPNet underperform SpSA as a GMRES preconditioner in many cases? Can you characterize the regimes where RAPNet is better (problem type, anisotropy/heterogeneity, graph structure, target sparsity), and provide guidance for when to choose it?
2. How robust is patch-trained generalization when moving to very different graphs/PDE families (distribution shift)? Any evidence beyond the reported benchmark set (e.g., cross-family transfer, different discretizations, stronger heterogeneity)?

**Limitations:**

The paper does not explicitly discuss limitations. Two points seem important.

First, RAPNet is not consistently strong in the most common AMG use case: as a preconditioner for Krylov solvers (e.g., GMRES). In many settings it lags behind SpSA, so the practical impact is less clear unless the method can be positioned in the regimes where it actually wins.

Second, RAPNet is still tightly coupled to the classical AMG machinery. It depends on standard unsmoothed aggregation (piecewise-constant) to build/initialize the multigrid hierarchy and define the coarse/fine structure. This means the method is not a fully end-to-end replacement of AMG, and its success inherits a lot from the traditional pipeline; when that initialization is suboptimal, RAPNet may be limited as well.

**Strengths And Weaknesses:**

**Strengths**

1. RAPNet is quite flexible in practice: it is trained once on small local patches, but can be applied to large systems (up to million-node scale) and reused across multiple right-hand sides.
2. The evaluation covers a fairly broad set of benchmarks, and the problem sizes are non-trivial, which makes the results more convincing from an HPC perspective.
3. The paper does a solid job on the AMG background/motivation, and the visualizations help explain what the learned coarse operators are doing.

**Weaknesses**

1. RAPNet still relies on the standard AMG pipeline—specifically piecewise-constant unsmoothed aggregation—to initialize the hierarchy. Since this is the core ingredient that makes AMG work in the first place, it’s not too surprising that RAPNet behaves reasonably on typical benchmarks; the paper should be clearer about what is genuinely learned vs. what is inherited from the classical setup.
2. As a standalone solver component, RAPNet looks strong. But as a **preconditioner for GMRES**—arguably the main use case for AMG—the results are often worse than SpSA on many problems. This weakens the practical story unless the authors can explain when/why RAPNet should be preferred.

---

> ### Author Rebuttal · Authors · 2026-03-30
>
> We are sincerely grateful to the reviewer for the thoughtful and highly insightful critique of our manuscript. Your detailed comments demonstrate a deep technical engagement with the core challenges of AMG and have been crucial in helping us clarify the unique positioning and practical value of RAPNet. We appreciate your recognition of RAPNet's strength as a standalone solver component and the specific questions you raised concerning its performance as a GMRES preconditioner. We are confident that the following clarifications and planned revisions fully address your concerns and highlight where RAPNet offers a distinct and superior alternative to methods like SpSA and merit an “Accept” recommendation.
> # Weaknesses
> 1. RAPNet is explicitly designed to augment the existing solver with learned sparse corrections only. It does not execute during the actual solve. We will revise the manuscript to explain more clearly which parts are learned (the augmentation network executed during solver setup only) and which are inherited (the classical sparsity patterns). Learned pooling could be interesting for future research as well, but it is beyond the scope of this work. We note that this setup has advantages. For example, since RAPNet is a classical solver in inference, it obeys the existing non-Galerkin AMG theory. It is known [Schroder, 2010] that the prolongation matrices themselves are generally effective at approximating smooth error corrections, but it is the Galerkin projection that makes the process ineffective. That is also the reason why K-Cycles were invented (works by Ivan Notay). There is evidence (theoretical and empirical) that, in some problems, finding the best coarse operator can significantly improve the solver, but this is widely unexplored. In this work, we aim to leverage the power of neural networks for the task, and to the best of our knowledge, we are the first to do so.
> 2. We acknowledge that SpSA achieves fewer GMRES iterations on some benchmarks. Note that in most experiments, RAPNet does not fall much shorter of SpSA. Furthermore, a complete assessment of practical utility must pivot from raw iteration count to the overall time-to-solution and resource constraints. As shown in Table 2, SpSA materializes the full smoothed coarse operator, making its memory footprint and **setup computation potentially prohibitive**, as evident in the temporal Barabasi-Albert example. RAPNet, by preserving the minimal unsmoothed sparsity pattern, offers a faster and less memory-intensive setup (Table 2). Therefore, RAPNet should be preferred when time-to-solution is critical, as it can achieve the final solution faster in wall-clock time even if it requires a marginally higher number of GMRES iterations. Note that the iteration count gap between RAPNet and SpSA is small in most of our GMRES experiments.
> # Questions
> 1. We believe that much of the gap between SpSA and RAPNet is because RAPNet is currently trained as a standalone solver. Other training schemes, such as training under GMRES residuals or learning from SpSA, could be interesting to attempt in the future. Note that in most of our GMRES experiments, the gap between RAPNet and SpSA is quite small and in practice, RAPNet's setup takes less time than SpSA for larger and more highly-connected matrices, as shown in Table 2. Furthermore, SpSA executes on CPU only and nearly serially. We note this in the paper, but will revise the text to make sure these points are made clearer to the reader.
> 2. Like any learning-based method, we do not expect RAPNet to perform well when presented with inputs well outside its learned distribution. Some limited form of dimensional generalization is shown in the paper. Attempting to learn from multiple PDE or graph classes at once could be a compelling future direction, perhaps employing a variant of the RAPNet architecture with more parameter capacity. Our focus in the current paper is on showing the feasibility and empirical validity of the RAPNet approach.
>
> # Limitations
> RAPNet's design choice is to augment, rather than replace, a classical AMG hierarchy, meaning it cannot "save" a defective classical input. This is a deliberate contrast to end-to-end neural surrogates or neural operators. As the paper demonstrates, this approach allows RAPNet to scale effectively to significantly larger systems and deeper multigrid hierarchies than current end-to-end methods can handle.
>
> A dedicated Limitations section will be incorporated into the camera-ready version to discuss all limitations raised by the reviewers.
>
> For a discussion comparing SpSA and RAPNet performance under GMRES, please refer to our response to question 1.

---

> > ### Author Rebuttal · Reviewer_QZQC · 2026-04-02
> >
> > Thanks for the rebuttal. It clarifies RAPNet’s positioning as a **setup-time augmentation** of a classical AMG hierarchy (not an end-to-end learned solver), and the learned vs. inherited components are now clearer.
> >
> > However, I do not think this warrants a score increase. My main concern remains: as a **GMRES preconditioner** (the dominant AMG use case), RAPNet is often weaker than SpSA. The rebuttal argues we should focus on **time-to-solution/memory** rather than iteration counts, but does not provide a sufficiently systematic end-to-end comparison across benchmarks/regimes (setup+solve time, peak memory) to substantiate that claim beyond limited examples.

---

> > > ### Author Response · Authors · 2026-04-07
> > >
> > > We appreciate the reviewer's engagement on the critical issue of GMRES preconditioning. We want to clarify the narrative surrounding RAPNet's performance relative to SpSA.
> > >
> > > The statement that RAPNet is "often weaker" than SpSA as a GMRES preconditioner is inaccurate based on our full empirical results. **RAPNet is demonstrably superior or comparable to SpSA in most of our GMRES experiments.** SpSA primarily outperforms RAPNet only on the regularly structured 2D FEM PDE benchmarks, where other AMG variants (e.g., smoothed aggregation, classical AMG) are known to work relatively well.
> > >
> > > To our discussion, we list the average time-to-solution (TTS) for RAPNet and SpSA under GMRES for the largest experiments in the paper, averaged over 32 examples:
> > >
> > > |||SpSA time (s)|RAPNet time (s)
> > > ---|---|---|---
> > > Geo 2D|1M|3.3 $\pm$ 0.26|2.12 $\pm$ 0.04
> > > Geo 3D|512K|4.67 $\pm$ 0.53|2.02 $\pm$ 0.75
> > > WS|1M|7.06 $\pm$ 3.98|2.21 $\pm$ 0.27
> > > TBA|600K|84.19 $\pm$ 6.7|2.21 $\pm$ 0.22
> > > Aniso Diff|$1024^2$|7.94 $\pm$ 1.0|9.28 $\pm$ 1.08
> > > Adv-Diff|$1024^2$|2.55 $\pm$ 0.3|3.57 $\pm$ 1.13
> > >
> > > A core objective is to overcome SpSA's limitations when applied to graphs with highly-connected neighborhoods, such as those arising from complex network models (e.g., TBA, WS). SpSA's inherent reliance on CPU processing (even with thread-based parallelization) and the materialization of a denser, smoothed coarse operator make its setup time prohibitive for these graph structures.
> > >
> > > We stress that once the preconditioner is set up, the time per iteration is **identical** for RAPNet, SpSA, and standard AGG, as they all apply the same V-cycle to the **almost identical sparsity patterns**. The cycles differ mostly in the values of the operators. This means that for a similar iteration count, the setup time dominates.
> > >
> > > **Table 2** in the paper compares setup times. This clearly shows that SpSA's setup time dominates the overall TTS in some cases. In our largest TBA experiment, SpSA takes about **150 seconds**, while RAPNet takes only about **3 seconds**, which is **50x shorter** while including running the initial AGG and CPU-GPU data transfer time. This difference renders SpSA unusable for large, complex networks where fast turnaround is required, regardless of a marginally lower iteration count.
> > >
> > > To further this point, we created **random graphs with a few social hubs**, typical in social network graph settings. We took the random geometric graph and added 3 nodes that are each connected to 80% of a sub-population consisting of half of the nodes, meaning half of the graph is geometric, and the other half contains these 3 highly-connected nodes.
> > > These highly-connected nodes pass signals quickly across the diameter of the graph, making **iteration counts** small (with GMRES):
> > >
> > > ||AGG|SpSA|RAPNet
> > > ---|---|---|---
> > > 100K|14.6 $\pm$ 1.71|17.6 $\pm$ 8.32|10.5 $\pm$ 1.08
> > > 200K|13.0 $\pm$ 2.91|12.3 $\pm$ 4.95|9.6 $\pm$ 1.07
> > > 300K|13.1 $\pm$ 2.64|26.6 $\pm$ 20.72|8.9 $\pm$ 0.74
> > >
> > > In this case, SpSA’s setup scales **quadratically** instead of linearly, which is one of our motivations for this work. We list the average setup time, averaged over 10 examples. It clearly shows SpSA’s setup **completely dominates the TTS**:
> > >
> > > ||AGG setup (s)|SpSA setup (s)|RAPNet setup (s)
> > > ---|---|---|---
> > > 100K|0.15 $\pm$ 0|44.29 $\pm$ 2.61|0.3 $\pm$ 0.2
> > > 200K|0.31 $\pm$ 0|221.80 $\pm$ 8.59|0.48 $\pm$ 0.01
> > > 300K|0.49 $\pm$ 0|552.85 $\pm$ 14.82|0.76 $\pm$ 0.01
> > >
> > > **For a 400K node graph, SpSA’s setup took over 48 minutes**. These results again highlight the long setup time of SpSA on highly-connected neighborhood graphs.
> > >
> > > **In summary, RAPNet is not positioned to win every benchmark, but provides a fast, GPU-friendly sparsification method that scales effectively, especially for highly unstructured problems.** While SpSA currently achieves better iteration counts on 2D advection-diffusion, RAPNet brings the sparsification paradigm to modern GPU architectures, demonstrating better wall-clock time on highly unstructured graphs (TBA and WS), which are poorly served by existing methods. Our method provides a practical alternative for large-scale graph-based problems where the classical AMG setup remains dominant. It is true that in some cases, simple aggregation strikes a good balance between time and iterations, but AGG is known to be insufficient for many applications.
> > >
> > > We hope these results satisfy your main concerns. If so, please consider increasing your score. We will add these results to the paper if accepted.

---

### Official Review · Reviewer_zAHD · 2026-03-10

**Soundness:** 3
**Presentation:** 3
**Significance:** 3
**Originality:** 3
**Overall Recommendation:** 5
**Confidence:** 2

**Summary:**

The authors propose RAPNet, a graph neural network that learns sparse additive corrections to algebraic multigrid operators constructed by unsmoothed aggregation.
The GNN processes the AMG hierarchy as a sequence of composite graphs (one per level pair), predicting corrections to prolongation, restriction, and coarse-grid operators that are constrained to the sparsity pattern of the aggregation.
Inference occurs only during the solver setup phase, amortizing the cost over multiple solves while retaining the efficiency of unsmoothed-aggregation AMG.
Training on small subgraph patches enables generalization over entire classes of matrices.

Experiments span several toy examples:
geometric graphs generated by Delaunay triangulations of random points, Watts-Strogatz graphs, temporal Barabási-Albert graphs, and FEM discretizations of anisotropic and advection-diffusion PDEs.
Performance is measured by iteration count to reach a relative residual of $10^{-6}$, both as a standalone solver and as a preconditioner to GMRES.
RAPNet consistently reduces iteration counts relative to standard aggregation and matches or outperforms SpSA.

**Compliance With Llm Reviewing Policy:**

Affirmed.

**Final Justification:**

The RAPNet method is well motivated and the paper is well written. While a proof on convergence would have strengthened the paper, I understand that such guarantees are less common in numerical methods that incorporate learning. All my other concerns and questions have been adequately addressed in the rebuttal. With the promised changes (e.g., real-world 3D mesh experiments, ablation experiments, reported failure rates, limitations section, minor clarifications), I therefore raise my score to "Accept".

**Key Questions For Authors:**

- **Q1.** Ground-truth solution vectors are sampled from $\mathcal{N}(0, I)$ during training.
  Does this impose implicit assumptions on the structure of $A$, and how sensitive is performance to this distributional choice?
- **Q2.** Only two Jacobi iterations approximate the coarse solve during training (line 297, also see W4).
  Have the authors experimented with implicit differentiation (e.g., Blondel et al., 2022)?
- **Q3.** The hidden states are updated via projection matrices $\mathbf{W}_E$ and $\mathbf{W}_V$.
  Are these projections learned, or fixed?
- **Q4.** The SpSA baseline shows very high variance in Table 1, as standard deviations often exceed the mean (e.g., $513 \pm 412$, $303 \pm 405$ on Watts-Strogatz 128k).
  Does SpSA have hyperparameters, and if so, how were they tuned?
  Reporting failure rates (i.e., the fraction of instances reaching the 1,000-iteration cap) would make the comparison more informative.

**Limitations:**

In the supplement, on line 747, the authors note that a separate model is trained per dataset type.
This limitation to generalization should be discussed in the main body of the paper.

**Strengths And Weaknesses:**

### Strengths

- **S1.** The method is well motivated.
  The corrected operators are sparse by construction, inference is a one-time setup cost amortized over multiple solves, and the model generalizes across entire classes of matrices.
- **S2.** The related work section does a thorough job of placing the contribution in the landscape of learned solvers, neural operators, and classical sparsification methods.
- **S3.** The dimensional generalization experiment in Figure 6 is convincing.
  A model trained exclusively on 2D advection-diffusion transfers to a 3D problem and maintains favorable convergence properties.
  This strengthens the cost-amortization story and suggests the learned corrections capture structural operator properties rather than overfitting to a particular discretization.
- **S4.** The contributions of the three learned operator corrections are ablated in Table 3.

### Weaknesses

Soundness:
- **W1.** No convergence guarantees are provided.
  Some theoretical grounding would substantially strengthen the contribution.

Significance:
- **W2.** The experiments rely on synthetic benchmarks (random Delaunay triangulations, Watts-Strogatz graphs, Barabási-Albert graphs).
  For the 3D geometric case, evaluations on real meshes would be more convincing than triangulations on randomly sampled points.
- **W3.** In Table 1, SpSA outperforms RAPNet as a GMRES preconditioner on several benchmarks (e.g., 2D geometric, Watts-Strogatz, TBA). These negative results are not discussed.
- **W4.** The coarse solve during training is approximated by two Jacobi iterations (line 297), described only as a stability heuristic.
  The same approach is taken during evaluation (line 402).
  No justification is given for why two iterations suffice, and no ablation over this choice is provided.

Presentation:
- **W5.** The AMG background (Section 3) is hard to follow.
  $\Delta A_c$ is one of the learned quantities but appears without introduction in Eq. (2).
  The distinctions between $A_c$, $A_g$, and $A_l$ need to be made clearer.
  A similar issue occurs with $\mathbf{e}_f$, which is used but not defined. Differences between $\mathbf{e}_f$, $\mathbf{e}_c$, and $\mathbf{e}$ should be clarified.
- **W6.** In Section 4.1, Figure 1 is discussed before the notation it relies on is introduced ($\mathbf{F}_l^V$, $\mathbf{F}_l^E$, $\mathbf{H}_l^V$, $\mathbf{H}_l^E$).
  Reordering the figure reference or the definitions would improve readability.

---

> ### Author Rebuttal · Authors · 2026-03-30
>
> We are grateful to the reviewer for their thorough reading of our manuscript. We appreciate the constructive nature of your feedback, which demonstrates a strong technical engagement with RAPNet. We are confident that the revisions outlined below will substantially strengthen the paper and justify an “Accept” decision.
> # Weaknesses
> 1. We agree that convergence guarantees would strengthen the contribution. Non-Galerkin AMG theory provides guarantees only for small corrections. Since RAPNet is a classical solver in inference, it obeys the same theory. Beyond that, proving that a neural network learns what it needs or can is not common in ML, especially in a self-supervised learning setup like ours. Since RAPNet is a learned method, we cannot guarantee it will produce good corrections in all cases. We will explicitly state this limitation in a new Limitations section.
> 2. We acknowledge that incorporating evaluations on real-world 3D meshes would significantly strengthen our 3D geometric case. We will add 3D FEM experiments to Table 1 as well, like the example presented in Figure 6. Our initial focus was on establishing the fundamental viability and scalability of RAPNet across various synthetic problem types. However, we agree that demonstrating performance on complex, real-world data is essential. We note that the area where this work is most needed is the TBA example, which is highly unstructured and non-local. In this example, other AMG variants (smoothed aggregation, classical AMG) generate quite dense hierarchies in practice. In our paper, this can be seen in the long setup times of SpSA in Table 2, because of the dense Galerkin products.
> 3. We acknowledge that SpSA achieves fewer GMRES iterations on some benchmarks. Note that on most benchmarks, RAPNet does not fall much shorter than SpSA. Furthermore, a complete assessment of practical utility must pivot from raw iteration count to the overall time-to-solution and resource constraints. As shown in Table 2, SpSA materializes the full smoothed coarse operator, making its memory footprint and setup computation potentially prohibitive, especially in highly unstructured examples like the Temporal Barabasi-Albert graph. RAPNet, by preserving the minimal unsmoothed sparsity pattern, offers a faster and less memory-intensive setup (Table 2). Therefore, RAPNet should be preferred when time-to-solution is critical, as it can achieve the final solution faster in wall-clock time even if it requires a marginally higher number of GMRES iterations. Note that the iteration count gap between RAPNet and SpSA is small in most of our GMRES experiments, and RAPNet is generally the preferred option in the non-PDE examples, which are more unstructured.
> 4. In our experimentation, we observed that two iterations of Jacobi sufficed to achieve good results with RAPNet. We will add an ablation experiment to test the effects of different choices of Jacobi iterations in the coarse solve.
> 5. We appreciate the reviewer’s comments on the clarity of the mathematical background. We will revise the text to take these comments into account.
> 6. We will reorder the text to reference Figure 1 after introducing the notation used in the figure.
> # Questions
> 1. Since the ground-truth vectors are smoothed with the system operator $A$, they become algebraically smooth with respect to $A$, the more iterations are applied. With more iterations, they become algebraically smooth vectors for $A$ regardless of the initial distribution. The only requirement is that the ground-truth solution does not contain a component of the null-space of $A$, which we enforce by multiplying it by $A$ (i.e., $Ax$) before using it to generate data.
> This implies that the sampling method does not impose any structure implicitly.
> 2. While we have not experimented with implicit differentiation (e.g., Blondel et al., 2022), this could be an interesting direction for further research on RAPNet in the future. We thank the reviewer for this suggestion.
> 3. The weight matrices given by $W_E$ and $W_V$ are learned and are part of the set of parameters for RAPNet. We will clarify this in the paper.
> 4. SpSA does have hyperparameters. We have used standard hyperparameters used for methods like AGG and SpSA while not enabling unfair computational power in the form of significantly denser operators for SpSA. However, we did not perform an exhaustive search to find the best parameters for SpSA. Such a search would have to be accounted for in the number of iterations charged to SpSA. RAPNet, on the other hand, learns to output corrections and does not have any hyperparameters to tune during inference.
> We will add the failure rates to the paper. The failures are indeed the primary cause of the high variance in some experiments.
> # Limitations
> We will add a Limitations section detailing the dataset limitation noted by the reviewer and other limitations. We appreciate the reviewer’s attention to detail and their help in improving our paper.

---

> > ### Author Rebuttal · Reviewer_zAHD · 2026-04-02
> >
> > I thank the authors for their detailed response. All my questions and stated weaknesses have been addressed in their rebuttal. I agree with the authors that a formal proof of convergence guarantees is not necessary for this submission. With the promised changes (e.g., real-world 3D mesh experiment, ablation experiments, reported failure rates, limitations section, minor clarifications), I believe this will be a solid contribution to the literature.

---

> > > ### Author Response · Authors · 2026-04-06
> > >
> > > We are grateful for your feedback and your positive assessment of our planned revisions. We are particularly pleased that you feel all your initial questions and weaknesses have been addressed.
> > >
> > > We are committed to incorporating all the agreed-upon changes. Given your satisfaction with our response and your encouraging sentiment that the revised paper will be a strong submission, we respectfully ask that you consider raising your score. This support would significantly help us bring the contributions of RAPNet to the broader scientific community at ICML.
> > >
> > > Thank you once again for your constructive engagement and time.

---

### Official Review · Reviewer_Us53 · 2026-03-11

**Soundness:** 2
**Presentation:** 3
**Significance:** 3
**Originality:** 3
**Overall Recommendation:** 3
**Confidence:** 3

**Summary:**

This paper proposes RAPNet, a GNN-based learning framework designed to accelerate AMG methods for solving large sparse linear systems. Unlike prior approaches that either replace the solver entirely or require inference at every iteration, RAPNet operates exclusively during the setup phase. The core contributions of this work include a level-wise training curriculum, subgraph training with appropriate boundary conditions, and a setup-only inference strategy. The paper evaluates RAPNet on a range of graph Laplacian and PDE-derived benchmarks, comparing it with two primary baselines: AGG and SpSA. Experimental results show that RAPNet significantly reduces the number of iterations on various problems while remaining highly competitive. Additionally, ablation studies validate the effectiveness of each core component in RAPNet.

**Compliance With Llm Reviewing Policy:**

Affirmed.

**Final Justification:**

The author has addressed all concerns. Enhancing the discussion of motivation and background in the introduction would further improve the overall logical quality of the article.

**Key Questions For Authors:**

1. RAPNet employs a composite graph architecture that enables cross-level message passing. If message passing are restricted strictly to adjacent levels (i.e., a standard hierarchical GNN), would you expect a significant degradation in performance?
2. The manuscript states that the number of V-cycles k used during training follows a uniform distribution U(1, 30). What empirical observations or theoretical considerations motivated the selection of this specific range?
3. What specific role does the sparse correction Δ P and Δ R learned by RAPNet play in algebra?
4. The experimental results demonstrate that RAPNet performs well overall. However, it remains unclear whether there are specific types of matrices or problem classes during the initial exploration that caused RAPNet to fail or converge poorly.
5. As noted in the experimental section (Lines 401–409), to maintain consistency with the training process, RAPNet employs two Jacobi iterations on the coarsest grid, whereas the baselines (AGG and SpSA) utilize an accurate direct solver. If the baselines are also restricted to two Jacobi iterations, would RAPNet's performance advantage remain as significant?

**Limitations:**

1. The performance of this method depends on the similarity between the training distribution and the testing distribution. It is still unclear how robust RAPNet is when faced with vastly different physical states that are not represented in the training set.
2. The manuscript exhibits inconsistencies in punctuation formatting (specifically quotation marks and dashes) and the reference list style. Addressing these inconsistencies is necessary to enhance the professionalism and readability of the paper.

**Strengths And Weaknesses:**

Strengths:
1. The paper is well-organized. The figures are clear, and their corresponding captions and descriptions are detailed.
2. The experimental evaluation settings are sound. In particular, the experiments cover a diverse set of problem scenarios; the comparisons with representative baselines are fair and rigorous, and comprehensive ablation studies further validate the contribution of each architectural component.
3. The paper demonstrates a deep understanding of numerical linear algebra. Notably, Section 3 provides a solid mathematical foundation for the development of subsequent methods.
4. The proposed method demonstrates practical effectiveness and scalability. Specifically, Table 2 clearly shows that RAPNet's setup time is negligible compared to classical AGG and significantly faster than SpSA on high-connectivity graphs.

Weaknesses:
1. The author summarized the three main contributions in a concise and clear manner on page 2 of the manuscript. However, in the introduction section, except for the third contribution which has a clear motivation explanation, the first two contributions lack sufficient background and motivation explanation, which makes their necessity appear insufficient.
2. While the paper presents excellent experimental results, it lacks a discussion on the impact of key hyperparameters on performance. A sensitivity analysis would be beneficial. Furthermore, the comparisons on lines 382–383 are limited to 'standard aggregation-based AMG (AGG)' and 'SpSA (2015).' The absence of benchmarks against more recent state-of-the-art approaches weakens the claim of the proposed method's superiority.
3. Although the solve phase is hardware independent, the setup phase relies on GPU acceleration for GNN inference (Table 2 indicates the use of an RTX 6000). In HPC environments where GPUs are unavailable during the setup phase or only CPU clusters are accessible, this dependency may limit the practicality of RAPNet.

---

> ### Author Rebuttal · Authors · 2026-03-30
>
> We are grateful that the reviewer recognized the paper's strengths, and the effectiveness of RAPNet's setup time. Your critiques have been helpful in identifying areas for improvement. We are confident that the proposed revisions address your concerns and warrant an "Accept" decision.
> # Weaknesses
> 1. Our training strategies will be clarified for a broader audience. We will explicitly frame these contributions as solutions to fundamental challenges in applying GNNs to multigrid.
> Namely, level-wise training ensures that RAPNet learns to optimize the components specific to each hierarchy level, leading to robust convergence across all scales.
> Subgraph training is motivated by scalability and generalization. By training on small localized subgraphs, RAPNet learns local rules for operator correction that transfer effectively to **much larger** unseen problems. This scaling generalization is shown in every experiment in Table 1.
> 2. More comprehensive testing against SOTA methods makes emerging ML methods more compelling. However, in this case, we could not find comparable ML-based methods: to the best of our knowledge, current SOTA methods are unable to scale past two MG levels, or are unable to scale to comparable problem sizes. In [Chen, ICLR 2025], they train a network per instance, whereas we train on the entire dataset.
> We used common defaults for classical method parameters and crucially, the AGG experiments and the inputs to RAPNet employ the exact same parameters. This ensures that the RAPNet experiments are strictly an improvement over the corresponding AGG experiment. Parameter values are detailed in the appendix.
> 3. Traditional HPC environments often do not include GPU hardware, instead relying on a large number of interconnected CPUs. In these cases, RAPNet inference can be quickly executed on CPU, since it is done only once at the initial setup and the network consists only of roughly 110K learnable parameters. These setup times for RAPNet evaluated on a consumer-grade CPU, averaged across 32 examples, will be added to the paper:
> |||Avg. Setup Time (s)
> ---|---|---
> TBA|400K|$9.24\pm0.21$
> TBA|600K|$11.13\pm0.29$
> TBA|800K|$16.37\pm2.65$
> WS|512K|$8.97\pm2.82$
> WS|1M|$14.48\pm1.92$
> AD|$512^2$|$12.77\pm2.29$
> AD|$1024^2$|$14.03\pm2.26$
>
> # Questions
> 1. In our initial experiments, we experimented with many different architectures, including those that consider each level in isolation or consider levels without inter-level message passing. Those architectures either performed poorly, failed to train or failed to make any meaningful impact on the resulting solver.
> 2. The 1-30 range for the initial iterations was chosen empirically, following our experimental observations where solvers started to stall as the vectors became smooth at about the 15-30 iteration mark. We agree that this range appears in the paper without sufficient motivation and will remedy this.
> 3. Our working hypothesis is that the corrections $\Delta P$ and $\Delta R$ serve the algebraic role of learned, localized re-weighting factors. Specifically, $\Delta P$ provides a scaling factor applied to the coarse-grid correction for each fine-node/aggregate combination independently. This learned correction may recover the convergence benefits associated with denser operators, while preserving the minimal footprint of the original AGG.
> We do not claim to have a rigorous explanation as to why the corrections to the transfer operators improve the solver. According to our ablation experiments, their impact is substantial.
> 4. During our experimentation, we did not find any problem instances where the initial solver is reasonable and yet the RAPNet-augmented solver fails. In some cases, problems do converge quickly enough that RAPNet has no leeway to improve performance, or the initial solver was too defective to be saved by RAPNet.
> 5. Depending on the problem, replacing the direct coarse solve with Jacobi iterations generally degrades the performance of the classical solvers. However, since a direct coarse solve is generally applicable to the problem sizes shown in the paper, we opted to show the best performance to avoid artificially degrading their performance. We expect RAPNet’s performance gap to remain the same or widen if Jacobi is used for the classical solvers.
> # Limitations
> 1. We did not attempt to test RAPNet’s ability to generalize well beyond its training distribution. Like any learning-based method, we do not expect it to perform well on data well beyond the training distribution. We did show one such case of generalizing from 2D to 3D (Figure 6), but we do not claim it is widely applicable.
> Beyond that case, RAPNet may be able to learn across more than one graph family at once, but we did not test this. The current paper’s focus is showing that the novel architecture is feasible.
> 2. We appreciate the reviewer’s attention to detail. We will fix all formatting issues, punctuation and reference inconsistencies.

---

> > ### Author Rebuttal · Reviewer_Us53 · 2026-04-02
> >
> > Thanks for the rebuttal. Is there any experimental data available to provide effective explanation for Question 1?

---

> > > ### Author Response · Authors · 2026-04-06
> > >
> > > The reviewer's inquiry regarding the necessity of cross-level message passing (Question 1) touches upon a core design principle of RAPNet as an improvement over **[Luz et al., 2020, ICML]**. Importantly, the **RAPNet architecture itself is not a hierarchical or composite GNN**. Rather, it is a message-passing GNN that operates on adjacent level-pairs of the composite multigrid graph. This input graph explicitly connects the adjacent AMG levels, using the transfer operators $P$ and $R$ to supply inter-level edges. This construction enables the cross-level message passing that the reviewer is asking about.
> > >
> > > In our initial architectural exploration, we experimented extensively with alternative designs, including those that restrict message passing, as suggested. Specifically, our experiments included:
> > > * Variants of Graph U-Net that utilize AMG’s $P$ and $R$ as pooling and unpooling operators.
> > > * Methods that consider the multigrid levels entirely in isolation, learning corrections independently for each level's transfer operators without any inter-level communication.
> > > * Incorporating the exact inverse of the coarsest grid into training. Since the coarse grid is an ill-conditioned matrix, i.e., where small changes can significantly change eigenvalues, this resulted in very unstable training, which failed most of the time.
> > >
> > > Notably, **none of these alternative architectures yielded any measurable improvement over the current state-of-the-art classical multigrid solvers (AGG, SpSA)**. They either failed to train stably, produced negligible impact on solver convergence, or yielded a non-convergent or stagnant solver.
> > > For this reason, we chose to pursue the paradigm established and proven to work in Luz et al. where Luz et al. was restricted to interpolation learning for symmetric Galerkin operators. We improved upon their approach by constructing and testing the multi-level solver vs. two-level only, changing the loss function, and propagating the features forward through each level-pair in the composite graph. This enabled training on small sub-graphs and inference on larger ones with more levels. In addition, we consider a broader collection of test problems, including challenging graphs and non-symmetric problems, which are considered to be much harder than symmetric ones. As evidence, note that **Table 3 presents an ablation experiment** where inter-level mixing (forward feature propagation) is disabled and RAPNet performs notably worse as a result.
> > >
> > > As is common practice in the machine learning literature, results from initial, unsuccessful architectural explorations are typically not reported. We do not claim that our choices are the best or that we have exhausted all architectural options, but at the same time, we can only report that these experiments failed for us, i.e., that they yielded non-convergent solvers or had negligible impact for small learning rates.

---

### Official Review · Reviewer_fru4 · 2026-03-12

**Soundness:** 3
**Presentation:** 3
**Significance:** 3
**Originality:** 3
**Overall Recommendation:** 4
**Confidence:** 4

**Summary:**

The paper proposes **RAPNet**, a graph neural network method for improving algebraic multigrid (AMG) by learning sparse corrections to the prolongation, restriction, and coarse-grid operators. Starting from an aggregation-based AMG hierarchy, the method keeps the sparsity pattern fixed and learns only the operator values. Neural inference is used only during setup, while the solve phase remains a standard sparse AMG iteration. Experiments on graph Laplacian and PDE problems show better convergence than classical aggregation and sparse-smoothed-aggregation baselines, with some generalization from small training graphs to larger test instances.

**Compliance With Llm Reviewing Policy:**

Affirmed.

**Final Justification:**

The authors provide detailed responses and more convincing experiments. Therefore, I raise my score to 4.

**Key Questions For Authors:**

1. Please explain more clearly, at the beginning of the paper, what **algebraically smooth error** means in AMG and why approximating it is the main role of the prolongation operator. A short introduction to the basic AMG principle would make the paper much more accessible.
2. In Table 1, NNZ seems to denote the number of nonzeros in the system matrix. Could the authors also report the **total hierarchy complexity**, for example, the total NNZ across all levels for AGG, SpSA, and RAPNet? Since sparsity is central to the paper, this would be very informative.

3. What is the size and cost of RAPNet itself? In particular, how many trainable parameters does it use, and what are its storage and runtime overheads relative to the AMG hierarchy it produces?

4. The paper argues that RAPNet can recover much of the benefit of denser coarse operators within a fixed sparse pattern. Can the authors provide more insight into when this is expected to work well, and when the fixed aggregation sparsity pattern may be too restrictive?

**Limitations:**

No. The paper mentions empirical performance and setup cost, but the limitations discussion should be more explicit. In particular, the manuscript should clearly acknowledge that (1) there is currently **no rigorous guarantee of spectral equivalence or multigrid convergence** for the learned sparse operators, (2) the experimental evaluation remains limited and does not yet include harder AMG benchmarks such as elasticity or Maxwell problems, and (3) the method improves only the coarse-grid hierarchy, while the smoother remains entirely classical.

**Strengths And Weaknesses:**

### Strengths

1. **Well-motivated idea.**
   The paper addresses an important AMG tradeoff: maintaining sparsity while improving coarse-grid quality. Learning corrections on top of a classical aggregation hierarchy is a reasonable and practically motivated design.

2. **Practical potential.**
   Using the neural network only in the setup phase is an attractive feature, since the solve phase remains a standard sparse AMG iteration without per-iteration inference. This makes the approach more plausible for repeated-solve settings and large-scale use.

3. **Promising empirical results.**
   The reported improvements over classical aggregation and sparse-smoothed-aggregation baselines are encouraging. The observed generalization from small training graphs to larger test problems is also a positive sign.

4. **Clear high-level message.**
   The main idea is easy to state: start from unsmoothed aggregation, preserve sparsity, and learn better operator values within that pattern. This is a clean and practically meaningful formulation.

### Weaknesses

1. **Presentation and accessibility.**
   The exposition is not yet sufficiently accessible to a broad ML audience. In particular, the paper should explain the basic AMG principle more clearly at the outset, especially what **algebraically smooth error** means and why the prolongation operator should approximate it. As written, readers without prior AMG background may have difficulty understanding the role of the learned corrections.

2. **Limited theoretical justification.**
   The paper argues that learned sparse corrections can recover much of the benefit of denser Galerkin or smoothed operators while preserving sparsity. However, unlike classical spectral sparsification or non-Galerkin AMG theory, it does not provide rigorous guarantees on spectral equivalence, approximation quality, or two-grid convergence. The justification is therefore largely heuristic, making the work more empirical than theoretical.

3. **Positioning relative to prior literature could be sharper.**
   There is substantial classical work on constructing sparse coarse operators with controlled approximation or spectral properties, including spectral sparsification methods such as the Spielman--Teng Laplacian solver. The paper should position itself more explicitly relative to this literature, especially for graph Laplacian problems. At present, the connection to prior theory is underdeveloped.

4. **Benchmark scope is limited.**
   The experiments are encouraging, but they are concentrated on graph Laplacians and a small set of PDE problems. The evaluation would be much stronger if it included more challenging AMG settings such as **linear elasticity** or **Maxwell-type problems**, where the near-null space is nontrivial and transfer operators are harder to construct. Such tests would better assess the generality of the method.

5. **The smoother remains classical.**
   AMG performance depends on both smoothing and coarse-grid correction. This work improves the hierarchy but leaves the smoother unchanged. That is a reasonable design choice, but it also limits the scope of the contribution. The paper should discuss this limitation more directly, including whether the smoother could also be learned or adapted.

---

> ### Author Rebuttal · Authors · 2026-03-30
>
> We thank the reviewer for their thoughtful feedback recognizing RAPNet as a well-motivated idea with practical potential. The constructive critiques have helped identify where the paper can be made more accessible and rigorous.
>
> # Weaknesses
> 1. Making AMG concepts accessible to a broader ML audience is important. We will revise the introduction to clearly define algebraically smooth error and explain the role of the operators, and improve the flow for readers unfamiliar with AMG principles. Note that the appendix includes a more comprehensive introduction to AMG.
> 2. We will revise the paper to make it explicit that we do not claim theoretical guarantees and that our claims are empirical. Additionally, we will add a new Limitations section (see below). While non-Galerkin AMG theory provides some guarantees, it is limited. The existing heuristics guarantee convergence only if the perturbations are small. Because RAPNet is a hybrid method, the solver remains an AMG iteration and will obey the theory as well if the corrections are small. In practice, we do not force the corrections to be small, similarly to SpSA. Like SpSA, RAPNet may stall or diverge. In addition, since RAPNet is an ML method, there is no guarantee it will produce good corrections in all cases. Generally, proving that a NN learns what it needs is not common in ML. This self-supervised regression problem, learned with random smooth test vectors, is a complicated setting from a theoretical ML perspective and not easy to analyze.
> 3. Positioning RAPNet relative to classical literature such as Spielman-Teng (ST) is important. We will discuss the ST solver in the camera-ready version. Similar to SpSA, classical algorithms like ST often rely on expensive serial graph algorithms that are challenging to apply in a GPU/LA computing paradigm, which limits their practicality for large-scale problems. In contrast, RAPNet leverages efficient GNN inference, providing a parallel, cost-effective approach.
> 4. More challenging settings, such as linear elasticity or Maxwell-type problems, which feature richer near-nullspaces, are important. While our initial evaluation established the viability of learning sparse corrections, we can include such systems in the camera-ready version. We believe that such problems should be tackled in combination with other techniques like block relaxations or block preconditioning. If accepted, we will show the use of RAPNet for Navier-Stokes or linear elasticity equations with scalar block preconditioning.
> 5. We agree that changing the smoother parameters can, in principle, improve the method. However, our experimentation with this idea has not been successful. Our understanding is that the smoothers should remain the ones that deal with high-frequency modes, and keeping them (with the updated operators and their diagonals) has been most effective for us. We believe that this is because our loss function mostly targets smooth error modes (reflected in the data), which are the ones for which the coarse-grid correction is most needed.
>
> # Questions
> 1. We agree that this information is not common knowledge in the ML community. We will define and explain algebraically smooth error in the introduction.
> 2. We confirm the total hierarchy complexity for RAPNet is identical to that of AGG. This is a key design feature: RAPNet strictly preserves the sparsity of the original aggregation. SpSA's complexity differs only slightly due to reweighting of some edges. We will state this explicitly.
> Generally speaking, we used common parameters for the classical methods, known to result in a complexity between 1.0-2.5.
> We will add the average complexities for the largest experiments to the paper:
> ||AGG/RN|SpSA
> ---|---|---
> 2D Geo|1.12|1.11
> 3D Geo|1.05|1.05
> WS|1.14|1.13
> TBA|1.71|1.65
> Aniso-Diff|1.26|1.23
> Adv-Diff|1.2|1.21
>
> 3. RAPNet is a weight-shared architecture, i.e., it contains a few trainable parameters, about 110K. This count is independent of the input size, making its footprint small relative to the inputs as they scale.
> The inference cost is incurred exactly once per input. We have detailed this cost in Table 2, showing that the overhead is well within practical limits for repeated-solve scenarios.
> This is a core strength: the initial cost to run does not depend on the number of right-hand sides.
> 4. RAPNet works best when the underlying aggregation pattern captures the necessary connectivity, but the values are suboptimal. We will discuss that on inherently "hard" graphs, the fixed aggregation sparsity pattern may be too restrictive.
>
> # Limitations
> We will add a new Limitations section to the final paper that explicitly addresses all points raised: RAPNet lacks rigorous theoretical guarantees; the experimental evaluation focuses on Laplacians and some PDEs; RAPNet focuses on improving the operators only, while the smoothers remain classical. We will discuss this as a scope limitation and an avenue for future research.

---

> > ### Author Rebuttal · Reviewer_fru4 · 2026-03-31
> >
> > **Weakness 4** Can the authors provide some preliminary tests on more challenging problems? The statements *We believe that such problems should be tackled in combination with other techniques like block relaxations or block preconditioning* conflicts with the reply in **Weakness 5** *However, our experimentation with this idea has not been successful* on testing better smoothers.
> >
> > Block relaxation can be a better smoother in some settings, such as anisotropic elliptic equations, where line smoothers improve performance.

---

> > > ### Author Response · Authors · 2026-04-06
> > >
> > > We thank the reviewer for the response. The statement for **Weakness 4**, as we demonstrate below, is not related to the statement for **Weakness 5**. Our response to **Weakness 5** was meant to explain why we did not include the relaxation parameters in the network, which includes only the Jacobi omega in this case. Our attempts to learn omega have not been successful. In our opinion, this is because it must be chosen such that the relaxation damps oscillatory modes. This is supported by multigrid theory. On the other hand, the coarse grid correction targets smooth modes, which our data generation procedure is designed to promote. Learning both the coarse grid and the relaxation damping parameters in one coupled learning system has not been successful, while treating them separately has been successful as shown in the paper.
> > >
> > > The statement in **Weakness 4** implies, for example, that cell-wise relaxation may be needed in the V-cycle. This can be used on the estimated coarse operators without learning, and its damping parameters can be estimated separately.
> > > The cell-wise method inverts each local submatrix for each block (cell) instead of just applying the point-wise update as in Jacobi. This solution is robust but is considered expensive, as it requires inverting $7\times7$ matrices for each cell in the case of a 3D problem at every iteration. In addition, the cell-wise relaxation is not easy to vectorize, and here, our framework also requires the derivatives of the relaxation. This requires a specialized, non-trivial implementation. On the other hand, most problems of interest can also be solved using block-preconditioning. Specifically, this applies to linear elasticity, where block-preconditioning runs much faster than the cell-wise relaxation method, according to [H. Wokber, 2010].
> > > In short, the aforementioned equations are systems of equations with a rich near-nullspace arising from the dominant grad-div operator i.e., Poisson ratio approaching 0.5, and standard iterative methods struggle. [Gaspar et al., 2008] states that by introducing a new pressure variable (the divergence of the displacement), the elasticity system can be transformed into a regularized saddle-point system
> > > $$\begin{bmatrix}A&B^\top\\\B&-C\end{bmatrix} \begin{bmatrix} \vec{u} \\\ p \end{bmatrix} = \begin{bmatrix} \vec q \\\ 0 \end{bmatrix}.$$
> > > This formulation is key to block-wise relaxation methods or block-preconditioning for elasticity, as it resembles the Stokes equation, where it was applied successfully as well. To form a preconditioner, we apply the pressure-Poisson approach, i.e. take the divergence of the first equation. This yields a Poisson equation for the pressure $p$. Consequently, the left-preconditioned error-residual system is
> > > $$ \begin{pmatrix} I&0\\\B&-A_p\end{pmatrix} \begin{pmatrix}A&B^\top\\\B&-C\end{pmatrix} \begin{pmatrix}\vec{e}_u\\\e_p \end{pmatrix}=\begin{pmatrix} I&0\\\B&-A_p\end{pmatrix}
> > > \begin{pmatrix}\vec r_u\\\r_p\end{pmatrix},$$
> > > where $-A_p=\mu BB^\top$ is the discrete Laplacian operator for $p$. Using the inexact commutativity of $BA\approx A_pB$, which holds in continuous space but not in discrete space, this system is equivalent to the following block-triangular system
> > > $$\begin{pmatrix} A&B^\top\\\0&BB^\top+A_pC\end{pmatrix} \begin{pmatrix}\vec{e}_u\\\e_p\end{pmatrix}= \begin{pmatrix} I&0\\\B&-A_p\end{pmatrix} \begin{pmatrix}\vec{r_u}\\\r_p\end{pmatrix}.$$
> > > The bottom-left zero block is only approximate, and hence the solution will take a number of iterations, even if the principal submatrices are inverted exactly. In practice, we approximate the inverse of the principal submatrices by applying one V-cycle. In summary, in each iteration, we apply one V-cycle for each scalar unknown (i.e., 3 V-cycles in 2D).
> > >
> > > For linear elasticity, all principal submatrices are standard discrete Laplacians. While our method is not intended to solve these problems in particular (geometric multigrid, or classical AMG are preferred), below we use V-cycle hierarchies that were generated by the network  trained on anisotropic diffusion problems. We believe that with more time, we could train a model to yield better hierarchies for this Poisson problem with our framework.
> > >
> > > The tables below summarize the number of iterations to convergence for the linear elasticity saddle-point formulation as a standalone solver and as a preconditioner to GMRES, respectively:
> > >
> > > Elasticity equation, $\\lambda=10$, $\\mu=1$
> > > ||AGG|SpSA|RAPNet
> > > ---|---|---|---
> > > $128^2$|202|Diverges|139
> > > $256^2$|421|Diverges|245
> > > $512^2$|425|Diverges|292
> > > $1024^2$|681|Diverges|384
> > >
> > > ||AGG|SpSA|RAPNet
> > > ---|---|---|---
> > > $128^2$|68|50|55
> > > $256^2$|128|65|102
> > > $512^2$|135|68|112
> > > $1024^2$|212|83|155
> > >
> > > Stokes equation
> > > ||AGG|SpSA|RAPNet
> > > ---|---|---|---
> > > $128^2$|393|Diverges|365
> > > $256^2$|394|Diverges|367
> > > $512^2$|374|Diverges|335
> > > $1024^2$|371|Diverges|330
> > >
> > > ||AGG|SpSA|RAPNet
> > > ---|---|---|---
> > > $128^2$|98|Stalls|103
> > > $256^2$|106|Stalls|104
> > > $512^2$|107|Stalls|138
> > > $1024^2$|104|Stalls|113

---

### Decision · Program_Chairs · 2026-04-30

**Decision:**

Accept (regular)

**Comment:**

The paper proposes a new scalable framework for acceleration of the algebraic multigrid by learning the correction. The reviewers have raised some concerns, but they have been mostly addressed, except one: the performance of the learned preconditioner is worse when used together with GMRES, it is not a very good sign. However, a  plain solver, it works well. Also, the changes provided in the rebutall needs to be integrated into the main text, since a lot of things have to be added. Overall, I am mildly positive over this paper.